# AI Agents That Matter

**Sayash Kapoor**[*]                                                                                     *sayashk@princeton.edu*
*Department of Computer Science*
*Center for Information Technology Policy*
*Princeton University*

**Benedikt Stroebl**[*]                                                                                  *stroebl@princeton.edu*
*Center for Information Technology Policy*
*Princeton University*

**Zachary S. Siegel**
*Department of Computer Science*
*Center for Information Technology Policy*
*Princeton University*

**Nitya Nadgir**
*Center for Information Technology Policy*
*Princeton University*

**Arvind Narayanan**                                                                                    *arvindn@cs.princeton.edu*
*Department of Computer Science*
*Center for Information Technology Policy*
*Princeton University*

**Reviewed on OpenReview:** *https://openreview.net/forum?id=Zy4uFzMviZ*

## Abstract

AI agents are an exciting new research direction, and agent development is driven by benchmarks. Our analysis of current agent benchmarks and evaluation practices reveals several shortcomings that hinder their usefulness in real-world applications. First, there is a narrow focus on accuracy without attention to other metrics. As a result, SOTA agents are needlessly complex and costly, and the community has reached mistaken conclusions about the sources of accuracy gains. Our focus on cost in addition to accuracy motivates the new goal of jointly optimizing the two metrics. We design and implement one such optimization, showing its potential to greatly reduce cost while maintaining accuracy. Second, the benchmarking needs of model and downstream developers have been conflated, making it hard to identify which agent would be best suited for a particular application. Third, many agent benchmarks have inadequate holdout sets, and sometimes none at all. This has led to agents that are fragile because they take shortcuts and overfit to the benchmark in various ways. We prescribe a principled framework for avoiding overfitting. Finally, there is a lack of standardization in evaluation practices, leading to a pervasive lack of reproducibility. We hope that the steps we introduce for addressing these shortcomings will spur the development of agents that are useful in the real world and not just accurate on benchmarks.

## 1 Introduction

Compound AI systems, or AI agents, are becoming an important research direction. Zaharia et al. (2024) argue that "compound AI systems will likely be the best way to maximize AI results in the future, and might be one of the most impactful trends in AI in 2024." Over a dozen agent benchmarks have been released, spanning domains such as web interaction (Zhou et al., 2024), programming (Jimenez et al., 2023) and tool

use (Ruan et al., 2024). Many benchmarks developed for LLM evaluation have also been used for agent evaluation.

Agent evaluation differs from language model evaluation in fundamental ways. Agents can be used on tasks that are harder, more realistic, have more real-world utility, and usually don't have a single correct answer. For example, agents can use the command line to carry out tasks; SWE-Agent even includes its own agent-computer interface (Yang et al., 2024). Agents can cost much more than a single model call. For example, the authors of SWE-Agent capped *each* run of the agent at $4 USD, which translates to hundreds of thousands of language model tokens.

As a result, agent benchmarking comes with distinct challenges. This paper empirically demonstrates these challenges and provides recommendations for addressing them. Specifically, we make five contributions.

1. **AI agent evaluations must be cost-controlled (Section 2).** The language models underlying most AI agents are stochastic. This means simply calling the underlying model multiple times can increase accuracy (Li et al., 2022; Chen et al., 2024; Li et al., 2024). We introduce three new simple baseline agents and empirically show that they outperform many SoTA complex agent architectures on HumanEval (Zhong et al., 2024; Zhou et al., 2023; Shinn et al., 2023) while costing much less. Therefore, agent evaluations must be cost-controlled; otherwise it will encourage researchers to develop extremely costly agents just to claim they topped the leaderboard.

2. **Jointly optimizing accuracy and cost can yield better agent design (Section 3).** Visualizing evaluation results as a Pareto curve of accuracy and inference cost opens up a new space of agent design: jointly optimizing the two metrics. We modify the DSPy framework (Khattab et al., 2023) for joint optimization, lowering cost while maintaining accuracy on HotPotQA (Yang et al., 2018).

3. **Model developers and downstream developers have distinct benchmarking needs (Section 4).** Through a case study of NovelQA (Wang et al., 2024a), we show how benchmarks meant for model evaluation can be misleading when used for downstream evaluation. We argue that downstream evaluation should account for dollar costs, rather than proxies for cost such as the number of model parameters.

4. **Agent benchmarks enable shortcuts (Section 5).** We show that many types of overfitting to agent benchmarks are possible. We identify 4 levels of generality of agents and argue that different types of hold-out samples are needed based on the desired level of generality. Without proper hold-outs, agent developers can take shortcuts, even unintentionally. We illustrate this with a case study of the WebArena benchmark (Zhou et al., 2024).

5. **Agent evaluations lack standardization and reproducibility (Section 6).** We found pervasive shortcomings in the reproducibility of WebArena and HumanEval evaluations (Table 7). These errors inflate accuracy estimates and lead to overoptimism about agent capabilities.

The overarching goal of our work is to stimulate the development of agents that are useful in the real world and not just accurate on benchmarks. (1) and (2) above do this by incorporating metrics beyond accuracy into agent evaluation and optimization; (4) and (5) do so by improving precision about what a benchmark aims to measures and ensuring that it actually measures that; and (3) does both.

## 1.1 What is an AI agent?

In traditional AI, agents are defined as entities that perceive and act upon their environment (Russell & Norvig, 1995). In the LLM era, the term is used in a narrower way (a thermostat would qualify as an agent under the traditional definition). Many researchers have tried to formalize the community's intuitive understanding of what constitutes an agent in the context of language-model-based systems. Many of them view it as a spectrum — sometimes denoted by the term 'agentic' (Ng, 2024) — rather than a binary definition of an agent. We agree with this perspective. Since there are already many definitions, we do not provide a new one, but rather identify the factors that cause an AI system to be considered more agentic according to existing definitions. We found three clusters of factors.

- **Environment and goals.** The more complex the environment — e.g. range of tasks and domains, multi-stakeholder, long time horizon, unexpected changes — the more AI systems operating in that environment are agentic (Shavit et al., 2023; Gabriel et al., 2024). Systems that pursue complex goals without being instructed on how to pursue the goal are more agentic (Shavit et al., 2023; Chan et al., 2023; Gabriel et al., 2024).

- **User interface and supervision.** AI systems that can be instructed in natural language and act autonomously on the user's behalf are more agentic (Gabriel et al., 2024). In particular, systems that require less user supervision are more agentic (Shavit et al., 2023; Chan et al., 2023; Gabriel et al., 2024). We discuss the user supervision aspect in more detail in Section 5.2.

- **System design:** Systems that use design patterns such as tool use (e.g., web search, programming) or planning (e.g., reflection, subgoal decomposition) are more agentic (Weng, 2023; Ng, 2024). Systems whose control flow is driven by an LLM, and hence dynamic, are more agentic (Weng, 2023; Chase, 2024).

## 2 AI agent evaluations must be cost-controlled

### 2.1 Maximizing accuracy can lead to unbounded cost

Calling language models repeatedly and taking a majority vote can lead to non-trivial increases in accuracy across benchmarks like GSM-8K, MATH, Chess, and MMLU (Li et al., 2024; Chen et al., 2024; Sun et al., 2024).

When the agent environment has easy signals to check if an answer is correct, repeatedly retrying can lead to even more compelling performance gains (Villalobos & Atkinson, 2023). Li et al. (2022) showed that the accuracy of AlphaCode increases from close to 0% zero-shot to over 15% with 1,000 retries and over 30% with a million retries (accuracy is measured by how often one of the top 10 answers generated by the model is correct). Thus, there is seemingly no limit to the amount of inference compute that can increase accuracy, and scaling inference compute has been shown to improve performance in various applications (Welleck et al., 2024; Brown et al., 2024). Coding competitions often include signals of correctness, such as test cases, which serve as verifiers to check if a given solution is correct. Agent developers can keep sampling from an underlying model until the solution passes the test cases. Our results below show that this is true for HumanEval.

### 2.2 Visualizing the accuracy-cost tradeoff using a Pareto curve

In the last year, many agents have been claimed to achieve state-of-the-art accuracy on coding tasks. But at what cost? To visualize the tradeoff, we re-evaluated the accuracy of three agents.

Specifically, we included agents from the HumanEval leaderboard on PapersWithCode that share their code publicly (Chen et al., 2021): LDB (Zhong et al., 2024), LATS (Zhou et al., 2023), and Reflexion (Shinn et al., 2023). [1] These agents rely on running the code generated by the model, and if it fails the test cases provided with the problem description, they try to debug the code (Zhong et al., 2024), look at alternative paths in the code generation process (Zhou et al., 2023), or "reflect" on why the model's outputs were incorrect before generating another solution (Shinn et al., 2023; Zhou et al., 2023; Zhong et al., 2024).

We also evaluated the cost and time requirements of running these agents. In addition, we calculated the accuracy, cost, and running time of a few simple baselines.

- **GPT-3.5** and **GPT-4** models (zero shot; no agent architecture)

---

[1] Reflexion is absent from the PapersWithCode leaderboard, but it has a reported accuracy of 91% (higher than any other agents with publicly available code apart from LDB and LATS), so we included it in our analysis. AgentCoder, listed as the top-performing agent, did not include a link to the code on the benchmark at the time of our analysis (late April 2024), nor did it include a link to the code in the paper, so we did not include it.

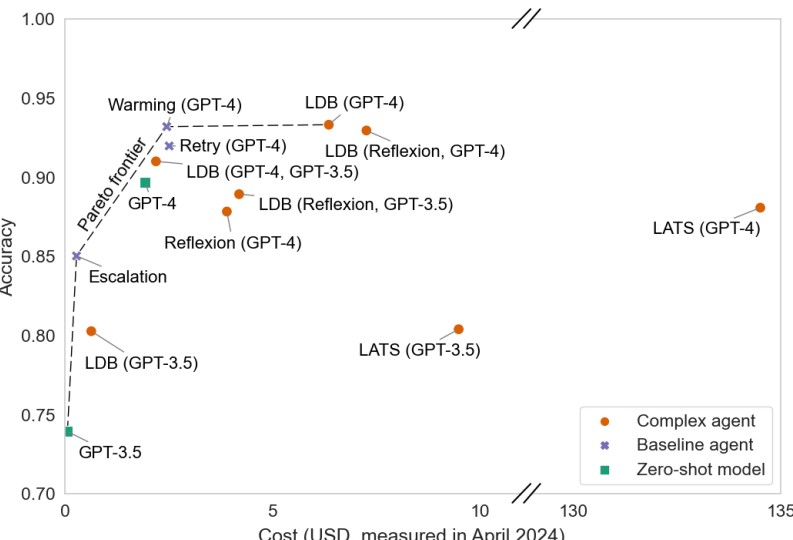

Figure 1: Our simple baselines offer Pareto improvements over SOTA agents. We run each agent five times and report the mean accuracy and the mean total cost on the 164 HumanEval problems. Where results for LDB have two models/agents in parenthesis, they indicate the language model or agent used to generate the code, followed by the language model used to debug the code. Where they have just one, they indicate that the same model was used to both generate the code and debug it. **Note the nonstandard axes**; In Appendix A, we show our results with the full y-axis as well as error bars and provide additional details. Robustness checks are contained in Appendix A.2. Section 4 explains why we measure dollar costs instead of using proxies for cost such as the amount of compute used.

- **Retry**: We repeatedly invoke a model with the temperature set to zero, up to five times, if it fails the test cases provided with the problem description. Retrying makes sense because LLMs aren't deterministic even at temperature zero (Appendix A.1).

- **Warming**: This is the same as the retry strategy, but we gradually increase the temperature of the underlying model with each run, from 0 to 0.5. This increases the stochasticity of the model and, we hope, increases the likelihood that at least one of the retries will succeed.

- **Escalation**: We start with a cheap model (Llama-3 8B) and escalate to more expensive models (GPT-3.5, Llama-3 70B, GPT-4) if we encounter a test case failure.

We use the modified benchmark version of HumanEval provided with the LDB paper (Zhong et al., 2024) since it includes example test cases for all 164 tasks (in the original benchmark, example test cases are provided for only 161 of 164 tasks, as detailed in Section 6).

## 2.3 Two-dimensional evaluation yields surprising insights

Fig. 1 shows our main results for this section. Note that an agent is on the Pareto frontier if there is no other agent that has significantly better performance on both dimensions simultaneously (see Appendix A.1). [2]

**"State-of-the-art" agent architectures for HumanEval do not outperform simple baselines**. There is no significant accuracy difference between our warming strategy and the best-performing agent architecture. In fact, we are not aware of any papers that compare their proposed agent architectures with any of the

---

[2]We constrain the Pareto frontier to be convex, because given two points $a$ and $b$ on the graph corresponding to agents A and B, we can always linearly interpolate between them by creating a new agent that invokes A with probability $p$ and B with probability $1 - p$. Hence, for instance, zero-shot GPT-4 is not on the frontier.

last three of our simple baselines on HumanEval (retry, warming, escalation).[3] This finding is supported by several other recent studies showing that scaling inference compute can increase accuracy (Brown et al., 2024; Hassid et al., 2024; Snell et al., 2024).

**Agents differ drastically in terms of cost**. For substantially similar accuracy, the cost can differ by almost two orders of magnitude. Yet, the cost of running these agents isn't a top-line metric reported in any of these papers. Reflexion and LDB cost over 50% more than the warming strategy, and LATS over 50 times more (all these costs are entirely or predominantly from calls to GPT-4, so these ratios will be stable even if model costs change). Meanwhile, the escalation strategy strictly improves accuracy while costing less than half of LDB (GPT-3.5).

**Lack of clarity on the source of performance gains.** There is a widespread belief in the AI community that complex ideas like planning, reflection, and debugging are responsible for accuracy gains on tasks such as HumanEval. However, based on our findings, the question of whether debugging, reflection, and other such "System 2" approaches (Kambhampati et al., 2024) are useful for code generation remains open, in line with other recent findings (Verma et al., 2024; Huang et al., 2023; Pan et al., 2024). The lack of clarity about the efficacy of System 2 approaches is exacerbated by a lack of reproducibility and standardization that we report in Section 6. Failing to identify the sources of empirical gains is a longstanding issue in ML and related fields (Lipton & Steinhardt, 2018; Henderson et al., 2018).

At the same time, models such as OpenAI's o1 have reported impressive gains owing to System 2 approaches (OpenAI, 2024a), though this involves training the model to be better at System 2 approaches using reinforcement learning (OpenAI, 2024b). Notably, o1 outperforms gpt-4o with retry and reflection even when accounting for inference compute (Epoch AI, 2024). Similarly, it is possible that System 2 techniques will be useful on harder programming tasks than those represented in HumanEval, such as SWE-bench (Jimenez et al., 2023).

To summarize this section, useful agent evaluations must control for cost — even if we ultimately don't care about cost and only about identifying innovative agent designs. Accuracy alone cannot identify progress because it can be improved by scientifically meaningless methods such as retrying.

## 3  Jointly optimizing cost and accuracy can yield better agent designs

Visualizing the cost and accuracy of agents as a Pareto frontier opens up a new space for agent design: jointly optimizing cost and accuracy, which can lead to agents that cost less while maintaining accuracy. This formalization is fully generalizable to other desiderata of agent design, such as latency.

The total cost of running an agent includes fixed and variable costs. Fixed costs are one-time expenses incurred when optimizing the agent's hyperparameters (temperature, prompt, etc.) for a given task. Variable costs are incurred each time the agent is run and depend on the number of input and output tokens. The more an agent is used, the more the variable cost dominates (Appendix B).

Joint optimization allows us to trade off the fixed and variable costs of running an agent. By spending more upfront on the one-time optimization of agent design, we can reduce the variable cost of running an agent (e.g., by finding shorter prompts and few-shot examples while maintaining accuracy).

As an illustration of the potential of joint optimization, we modify the DSPy framework (Khattab et al., 2023) and evaluate it on the HotPotQA benchmark (Yang et al., 2018). We chose HotPotQA because is one of the benchmarks used to illustrate the effectiveness of DSPy in the original paper and has been featured in several official tutorials by the developers. We used the Optuna hyperparameter optimization framework (Akiba et al., 2019) to search for few-shot examples to be included with an agent that minimizes cost while maintaining accuracy. Note that we expect more complex joint optimization approaches to vastly outperform our approach. Our results are only a starting point intended illustrate the vast, underexplored design space in agent design enabled by joint optimization.

---

[3]Chen et al. (2024) do compare the effect of retrying the same model multiple times on HumanEval, but they do not test the performance of GPT-4, nor do they argue for a specific complex agent architecture.

### 3.1 HotPotQA evaluation setup

We implement several agent designs to evaluate performance on multi-hop question-answering using DSPy. For retrieval, we use ColBERTv2 to query Wikipedia based on the HotPotQA task specification (Yang et al., 2018). Performance is evaluated by comparing whether the agent successfully retrieved all ground-truth documents that are part of the HotPotQA task. We use 100 samples from the HotPotQA training set to optimize the DSPy pipelines and 200 samples from the evaluation set to evaluate the results (this is consistent with the implementation of the DSPy pipelines provided by the developers to illustrate efficacy at multi-hop retrieval). We evaluate five agent architectures:

- **Uncompiled:** We do not optimize the agent's prompt or include instructions on formatting HotPotQA queries. Each prompt only contains the instructions for the task and the main content (i.e., question, context, reasoning) but no few-shot examples or formatting instructions.

- **Formatting instructions only:** This is the same as the uncompiled baseline, but we add instructions on how to format generated outputs for writing retrieval queries.

- **Few-shot:** We use DSPy to identify effective few-shot examples using all 100 samples from the training set. We include formatting instructions. Few-shot examples are selected based on successful predictions generated on the training set.

- **Random Search:** We use DSPy's random search optimizer on a subset of the training data (50 of 100 samples) to select the best few-shot examples based on its performance on the remaining 50 samples. We include formatting instructions.

- **Joint optimization:** We iterate over half the training set (50 of 100 samples) to collect a set of candidate few-shot examples that improve the model's accuracy. We use the other 50 samples for validation. We jointly maximize accuracy and minimize the number of tokens in the few-shot examples included in the prompt using parameter search. We implement parameter search using Optuna (Akiba et al., 2019). We search over the following parameters to find Pareto-optimal agent designs: (a) the temperature for each module within the agent, (b) the number of few-shot examples, (c) the selection of specific examples, and (d) whether to add formatting instructions. Of the candidate agents selected by Optuna, we pick the one with the best accuracy on the development set as our joint optimization model.

We test all five of the above agent designs on two underlying models: Llama-3-70B and GPT-3.5.

### 3.2 HotPotQA results: Joint optimization reduces cost while maintaining accuracy

Fig. 2 shows our main results. We confirm that DSPy offers substantial accuracy improvements over uncompiled baselines, but we find that this comes at a cost. Fortunately, we can mitigate the cost overhead — for GPT-3.5, joint optimization leads to 53% lower variable cost with similar accuracy compared to both default DSPy implementations. Similarly, for Llama-3-70B, it leads to a 41% lower cost while maintaining accuracy.

**Tradeoffs between fixed and variable costs for agent design.** Our joint optimization formulation provides a way to trade off fixed and variable costs (Appendix B). In particular, we find that if used for HotPotQA tasks, the Llama-3-70B as well as the GPT-3.5 joint optimization model both become cheaper (in terms of total cost) compared to the default DSPy implementation, after 1,350 tasks (Appendix B). For agents used in large-scale real-world tasks, the variable cost is by far the dominant term compared to the fixed cost, by orders of magnitude, as an agent might be used millions of times.

In summary, joint optimization allows for efficient agent design. This comes at a small fixed cost for optimizing the design, which is insignificant if the agent is used thousands of times.

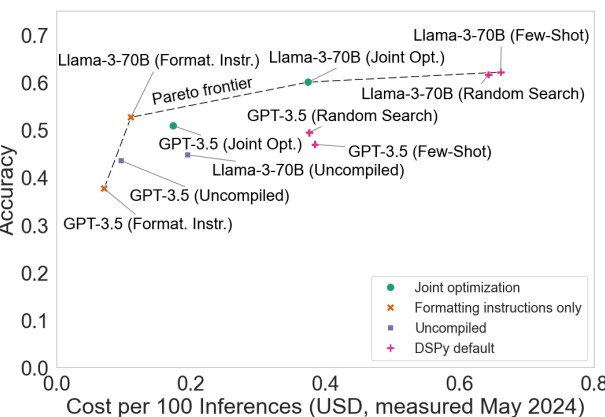

Figure 2: Joint optimization maintains accuracy while significantly reducing cost. All measurements are averages of 5 runs on the test set. A figure with error bars and other details about our empirical results are included in Appendix B. Note that DSPy has a much higher cost than our baseline agent designs since it always includes up to 8 few-shot examples per prompt (the number being a configurable parameter) without optimizing for cost, whereas the baseline agents include none, and joint optimization uses a variable number of examples.

## 4 Model and downstream developers have distinct benchmarking needs

AI evaluations are used by model developers and AI researchers to identify which changes to the training data and architecture improve accuracy (model evaluation) and also by downstream developers to decide which AI systems to use in their products for procurement decisions (downstream evaluation). The difference between model evaluation and downstream evaluation is underappreciated. This has led to much confusion about how to factor in the cost of running AI.

**Model evaluation** is a scientific question of interest to researchers. Here, it makes sense to stay away from dollar costs, because reporting costs breaks many properties of benchmarks that we take for granted: measurements don't change over time (whereas costs tend to come down) and different models compete on a level playing field (whereas some developers may benefit from economies of scale, leading to lower inference costs). Because of this, researchers usually pick a different axis for the Pareto curve, such as parameter count or training compute.

For model evaluation, controlling for compute is a reasonable approach: if we normalize training compute, we can then understand if factors like architectural changes or changes in the data composition are responsible for improvements, as opposed to more compute (Lambert, 2024).

**Downstream evaluation** is an engineering question that helps inform a procurement decision of which model or agent to use in a particular application. Here, cost is the actual construct of interest. The downsides of cost measurement in model evaluation are exactly what is needed for downstream evaluation. Namely, inference costs do come down over time, and that greatly matters to downstream developers. It is unnecessary and counterproductive for the evaluation to stay frozen in time.

**Proxies for cost are misleading for downstream evaluation.** In the context of downstream evaluation, proxies for cost (such as the number of active parameters or training compute) are misleading. For example, Mistral released a figure alongside their latest model, Mixtral 8x22B, to explain why developers should choose it over competitors (Mistral AI team, 2024). It used the number of active parameters as a proxy for cost. From the perspective of a downstream developer, this proxy is misleading. For example, as of June 2024, Mixtral 8x7B costs twice as much as Llama 2 13B on compute provider Anyscale. Yet Mistral's figure shows that it costs about the same because it only considers the number of active parameters.

Downstream developers don't care about the number of active parameters when they're using an API. They simply care about the dollar cost relative to accuracy. Mistral chose "active parameters" as a proxy, presumably because it makes their models look better than dense models such as Meta's Llama and Cohere's Command R+. If every model developer picked a proxy that makes their model look good, multi-dimensional evaluation would lose its usefulness.

**Addressing challenges to cost evaluation.** As discussed above, there are several hurdles to cost evaluation. Different providers can charge different amounts for the same model, the cost of an API call might change

overnight, and cost might vary based on model developer decisions, such as whether bulk API calls are charged differently.

These downsides can be partly addressed by making the evaluation results customizable using mechanisms to adjust the cost of running models, i.e., providing users the option to adjust the cost of input and output tokens for their provider of choice to recalculate the tradeoff between cost and accuracy. In turn, downstream evaluations of agents should include input/output token counts in addition to dollar costs, so that anyone looking at the evaluation in the future can instantly recalculate the cost using current prices. We have prototyped a simple example of such an interface.[4]

### 4.1 Implications for benchmark design using a case study of NovelQA

The difference between model and downstream evaluation can also lead to challenges in benchmark design. We show how such challenges arise with a case study of the NovelQA benchmark (Wang et al., 2024a). The benchmark is motivated by the need to evaluate language models with long context windows. Novel lengths in the benchmark range from 50,000 to over a million words. Each novel has between 5 and 100 questions about it. To evaluate an AI system on NovelQA, developers submit their model responses to benchmark questions to a centralized platform (CodaBench), and top-performing submissions are included in a public leaderboard.

This is a good benchmark for model evaluation, but it would be misleading if used for downstream evaluation by a developer looking to build a bot for answering questions about novels. Like the other benchmarks we have discussed, the NovelQA leaderboard does not measure cost. Nor is it easy to construct such a leaderboard. That's because NovelQA evaluates language models by asking *all* questions about a novel in one go, right after inputting the novel's content. However, this does not represent how users would ask questions about novels in practice. Even if users have many questions about a novel, they will likely ask them individually rather than all at once. Such sequential queries would cost orders of magnitude more because the novel has to be re-processed each time. In other words, the task of answering multiple questions about a novel (as implemented by NovelQA) doesn't tell us anything about the cost of asking questions sequentially.

In particular, when comparing the two main approaches to novel-based QA — long-context models and retrieval-augmented generation (RAG) — NovelQA makes RAG look much worse than it is in a real-world scenario. Specifically, we found that the two approaches are roughly equally accurate, with RAG costing more than 20 times less (Table 6). But on NovelQA, RAG costs half as much (a tenfold overestimate). As a result, the NovelQA leaderboard is misleading for downstream evaluation. To be clear, NovelQA authors don't claim that it is suitable for downstream evaluation. Still, in general it is a common practice for downstream developers to look to model evaluation benchmarks. Instead, we argue that downstream evaluation benchmarks must be separate from, or at least variants of, model evaluation benchmarks.

## 5 Agent benchmarks allow shortcuts

Benchmarks are useful if they give us an estimate of real-world accuracy. If a benchmark allows shortcuts, accuracy on the benchmark does not translate to the real world (McIntosh et al., 2024; Mialon et al., 2023; Wagstaff, 2012).

Overfitting is one prominent type of shortcut, and a serious problem for agent benchmarks, since they tend to be small — typically a few hundred samples. This is a much more serious problem than LLM training data contamination, as knowledge of test samples can be directly programmed into the agent as opposed to merely being exposed to them during training. In principle, a lookup table can achieve 100% accuracy on many agent benchmarks. Overfitting would be obvious to spot if an agent used a lookup table, but other types of overfitting can be much more subtle and hard to detect, which is of course why held-out test sets are crucial in machine learning. Yet, surprisingly, we find that many agent benchmarks do not include held-out test sets. In addition to creating a test set, benchmark developers should consider keeping it secret to prevent LLM contamination or agent overfitting.

---

[4]See: `https://benediktstroebl.github.io/agent-eval-webapp/`

But we can go further. What about overfitting to a task? For example, if an agent scores highly on a Python programming benchmark but cannot program in any other language, is it a problem? This depends entirely on the agent's purpose and what the benchmark creator desires in terms of the generality of the agent (Chollet, 2019; Morris et al., 2024). In our survey, we have found four levels of generality:

1. **Distribution-specific** benchmarks are limited to a specific task, such as U.S. grade school math problems, and do not account for distribution shifts.

2. **Task-specific** benchmarks are limited to a specific task such as booking a flight, placing an order on an e-commerce website (Yao et al., 2023), or solving a GitHub issue (Jimenez et al., 2023), and account for the possibility of distribution shifts, including drift. After all, an agent that can book flights today but breaks if the flight booking website changes its layout would not be very useful. Web agents have already been found to be sensitive to such small modifications to the GUI (Ma et al., 2024b).

3. **Domain-general benchmarks** aim to measure the ability to perform *any task in a specific domain*, such as web browsing or tool use (Zhou et al., 2024).

4. **General-purpose benchmarks** measure the accuracy of agents across different domains, such as the *same* agent being able to perform web browsing and robotics tasks. It is unclear if such benchmarks are necessary or if aggregating domain-general benchmarks can better serve the purpose.

We propose as a core principle that the greater the intended generality of the agent, the more the held-out set should differ from the training set, as detailed in Table 1. For example, if a benchmark is intended to be domain-general but doesn't contain held-out tasks, agent developers may (intentionally or unintentionally) take shortcuts that work only for the specific tasks represented in the dataset, resulting in agents that don't work well for other tasks in the domain. Benchmark developers must do their best to ensure that shortcuts are impossible. We view this as the responsibility of benchmark developers rather than agent developers, because designing benchmarks that don't allow shortcuts is much easier than checking *every single* agent to see if it takes shortcuts. Benchmarks that create a level playing field are the core reason for the rapid progress of ML over the last half century (Donoho, 2017; Simons Foundation, 2019).

Table 1: Appropriate holdouts based on level of generality. See Appendix C for details.

| Level of generality | What should be held out | Num. benchmarks with appropriate holdouts |
|---|---|:---:|
| Distribution-specific | In-distribution samples | 1 / 2 |
| Task-specific | Out-of-distribution samples | 4 / 10 |
| Domain-general | Tasks | 3/18 |
| Fully general | Domains | 0 / 3 |

There are many types of distribution shifts, and benchmark developers can't necessarily model all of them. But they must attempt to identify which distribution shifts are particularly likely for the task in question. Another approach — not always practical — is to evaluate *sim2real* transfer, where leading agents are evaluated not just on benchmark tasks but also the corresponding real-world tasks — for example, Amazon shopping for a web shopping benchmark Yao et al. (2023).

We analyzed 33 agent benchmarks and classified them into the four levels of generality (Table 5). Most are either task-specific or domain-general. In many cases, it wasn't clear which level of generality the benchmark developers had in mind, and we made our best guess based on how the benchmark was presented in the paper. This lack of clarity is problematic as it makes it hard to know what we can and can't conclude about the agents that perform well on that benchmark.

We recognize that time and resource constraints may hinder benchmark designers from creating a holdout set at the correct generality level. Hence we count a holdout as appropriate if there actually exists a holdout at the appropriate level of generality *or* the benchmark designers indicate an intent to create such a holdout.

However, as shown in Table 1 and Appendix C, the majority of benchmarks do not include an appropriate held-out set, including 17 that have no hold-out and no indication that a holdout set will be added in future editions of the benchmark.

Note that in traditional machine learning research, held-out test sets are usually at the level of in-distribution or out-of-distribution samples (the first two rows of the Table 1). This is sufficient because models are specific to a single task, say image classification or spam classification. But LLMs and domain-general agents are expected to handle tasks that are not known ahead of time and may be specified in natural language in some cases. This motivates the need for held-out tasks during evaluation.

### 5.1 Case study of the STeP agent on WebArena.

Web agents can be evaluated on many capabilities: navigating to a website, scrolling, selecting the right web element etc. There are many different types of websites that can be used such benchmarks: e-commerce, social media, information search etc. WebArena is an agent benchmark that aims to evaluate agents on tasks on the web (Zhou et al., 2024). It includes clones of six different websites (GitLab, Reddit, Wikipedia, OpenStreetMaps, an e-commerce platform, and a content management system) and two tools (calculator and scratchpad). It has 812 different tasks that involve interacting with these websites, such as "find the address of all US international airports that are within a driving distance of 60 km to the Niagara Falls" and "post a question on a subreddit related to New York City".

WebArena's core stated selling point seems to be realism, which means that it should be difficult to find shortcuts. If we consider WebArena a task-specific benchmark, the key type of distribution shift is drift: agents should be robust to changes made to a website over time. However, WebArena does not model drift. To be clear, it is challenging to model drift. Benchmark developers would need to find changes made to published websites and incorporate those into the test sets. Further, as we discuss above, the held-out set would need to be kept secret, since agent developers could otherwise overfit to the specific changes in the benchmark. Still, we view these steps as necessary for meaningful evaluation.

Consider the top agent on the WebArena leaderboard, called STeP (Sodhi et al., 2024). It has an accuracy of 35.8%, more than double the accuracy of the top-performing baseline agent introduced in the WebArena paper, and over 10 percentage points more than the next-best agent (Drouin et al., 2024). How does STeP achieve this high accuracy? It turns out that STeP hardcodes policies to solve the specific tasks included in WebArena. For example, several WebArena Reddit tasks involve navigating to a user's profile. The STeP policy for this task is to `look at the current base URL and add a suffix '/user/user_name'`. This is brittle: the policy would no longer be effective if the website updates its URL structure (an example of drift). Even if the probability of an individual policy failing is small, an agent might need to call different policies dozens of times for each task. The overall probability of failure compounds quickly.

To be clear, the STeP developers' goals are orthogonal to the benchmark developers' goals—creating composable policies for accomplishing fixed tasks that are known apriori. From this perspective, STeP's design choices make sense. Yet the leaderboard accuracy on WebArena (such as the accuracy of the STeP agent) is misleading from the perspective of downstream developers, who might be using the WebArena leaderboard to understand the accuracy of web agents on real-world tasks and make decisions about which agent to adopt in an application.

Things become even more problematic if we consider WebArena a domain-general benchmark. This can be justified based on the claim that for previous web benchmarks, "the functionality of many environments is a limited version of their real-world counterparts, leading to a lack of task diversity" (Zhou et al., 2024). This suggests that the WebArena developers aim to stimulate the development of agents that can accomplish many different tasks on the web.

Unfortunately, WebArena lacks a held-out test set for evaluating whether an agent can perform well on unseen web tasks. (It is hard to confirm if building a domain-general benchmark is indeed their main objective. This lack of clarity is problematic as it makes it hard to assess whether benchmark developers deliver on their promises and what the leaderboard accuracy truly reflects.) Note that if the held-out set contained different tasks (such as samples from completely new and unseen websites) compared to those in the training set,

the accuracy agents like STeP would be drastically lower, because none of the hardcoded policies would be effective.

## 5.2 Agent benchmarks don't account for humans in the loop

The degree of human supervision, feedback, and intervention required for an agent to perform a task can be seen as a spectrum. Consider a data analysis task. On one end of the spectrum, the analyst might use a chatbot to help with tasks like debugging. Here the user is firmly in control and verifies all chatbot outputs. Or the analyst might ask the agent to write and execute code for certain data visualization tasks. The analyst might not review every line of code but only intervene if something appears to be wrong. At the other extreme, an analyst might give the agent a dataset and a task (e.g. identify how the presence of a swimming pool impacts property value using a home sale database), giving it full autonomy over all aspects of the task such as how to perform causal inference.

Current evaluations predominantly focus on these two extremes: they either evaluate the capacity of chatbots to answer questions correctly (e.g., MMLU (Hendrycks et al., 2021)), or whether agents can perform a task with no supervision (e.g., the agent benchmarks in Table 5).

However, these evaluations do not reflect how people use chatbots and agents in the real world. For example, the humans using a chatbot might steer it towards the correct answer, give feedback on where it went wrong, or ask it to change something about the output. Similarly, an agent might have the capacity to take an action, but require user confirmation for consequential actions. For example, users read emails using ChatGPT's Zapier plugin by connecting their email accounts, but sending an email requires confirmation from the end user.

Human feedback might greatly increase accuracy. For example, Shi et al. (2024) find that simple feedback increases the performance of GPT-4 from 0% to over 86% on challenging programming problems—from entirely useless to almost perfect. Thus, the lack of human-in-the-loop evaluation of agents might lead us to underestimate their usefulness (whereas the lack of held-out test sets leads to overoptimism).

Of course, human-in-the-loop evaluation is costly and tricky. The agent's accuracy will depend not just on the system but also on the skill level of humans interacting with it. While addressing these concerns with human-in-the-loop evaluations is beyond the scope of this paper, it is an important direction for future research. Ibrahim et al. (2024) provide a detailed overview of implementing such evaluations in the context of safety evaluation.

## 6 Inadequate benchmark standardization leads to irreproducible agent evaluations

During our experiments, we identified several shortcomings in the reproducibility and standardization of agent benchmarks and evaluations. Our analysis is based on a widely accepted definition of reproducibility: that the code and data accompanying a paper should be enough to reproduce the results that it reports (National Academies of Sciences, Engineering, and Medicine, 2019). Without reproducible agent evaluation, it is hard to distinguish genuine improvements in agent designs from artifacts of the differences in evaluation choices. Since agents are often intended to be used by downstream developers, lack of reproducibility also misleads developers adopting agents in real-world applications. Finally, irreproducible results impose a huge time cost on researchers trying to build on claims of state-of-the-art results.

We identified five root causes for the lack of standardized and reproducible agent evaluations, all of which arise from the differences between LLM evaluation and agent evaluation. Since these issues relate to standardization, we view them as primarily the responsibility of benchmark developers rather than agent developers.

1. **Evaluation scripts make assumptions about agent design that aren't satisfied by all agents.** Many evaluation shortcomings result from the lack of a clear standard for releasing evaluation scripts alongside benchmarks (Biderman et al., 2024). Agent developers need to implement their own evaluation for their agents either because the evaluation script does not account for different agent designs or because the script provided by the benchmark developer itself has bugs. This leaves open the possibility for

non-standard evaluation scripts leading to incomparable results across agent developers evaluating their agents on the same benchmark.

2. **Repurposing LLM evaluation benchmarks for agent evaluation introduces inconsistencies.** In some cases, even if benchmark developers provide evaluation scripts, agent developers need to re-implement these scripts. This is because many prominent benchmarks have been designed for language model evaluation. Using these benchmarks to evaluate agents can require changes to the benchmark design and evaluation.

   For example, HumanEval does not include example test cases for 3 of the 164 problems. In addition, the example test cases are included in the docstring accompanying a problem, rather than being machine readable and easily extractable. Both of these design choices are suitable for evaluating language models, since models can be evaluated by prompting them with the included docstrings, even if there are no example test cases. But this benchmark design does not work for agents that rely on using the included example test cases and regenerate solutions if they are incorrect (such as LDB, Reflexion, LATS, and our simple baselines in Section 2). As a result, developers evaluate agents using varying subsets of benchmarks or adding example test cases to the original benchmark. For example, the Reflexion (Shinn et al., 2023) and LATS (Zhou et al., 2023) modified HumanEval by removing the problems without example tests. LDB (Zhong et al., 2024) added example tests to the 3 problems which did not originally include them and convert examples to a machine-readable format (we use the version provided by the authors of LDB). But agent and model evaluations are clubbed together on leaderboard websites such as PapersWithCode, despite the varying evaluation methods. Aggregators of benchmark results (such as PapersWithCode) must ensure that evaluations on the same leaderboard were all conducted using the same methods.

3. **The high cost of evaluating agents makes it is hard to estimate confidence intervals.** Agents might call the underlying language models hundreds (or thousands) of times. This means that evaluating agents can be extremely expensive. For example, SWE-bench (Jimenez et al., 2023) consists of over 2,000 programming tasks. The authors of SWE-Agent (Yang et al., 2024) set a cost limit of USD 4 per task. Evaluating SWE-Agent on the entire benchmark could cost over USD 8,000 *for a single evaluation run.* The high cost makes it infeasible to run evaluations multiple times, and perhaps as a result, agent evaluations are rarely accompanied by error bars. This makes it hard to understand the variance of reported results. We found that many reported accuracy scores were above the maximum of five runs that we performed in our reproduction attempts, and the reported baselines were in some cases lower than the minimum of five runs we performed.

4. **Agent evaluation relies on external factors such as interacting with an environment which can lead to subtle errors.** LLM benchmarks typically consist of input strings and rely on strings as outputs, whereas agents often involve dynamic interactions with environments such as the web or a command line, which are not easily reducible to static inputs and outputs. This can lead to incorrect assumptions. For example, one common assumption in evaluation is invariance to the order in which tasks in a benchmark are evaluated, as tasks are presumed to be independent. This assumption fails for WebArena (Zhou et al., 2024). One of the websites included in the benchmark (a clone of Reddit) has rate limits for certain agent actions, such as posting content. As a result, if the tasks involving Reddit posts are evaluated one after another, they are much more likely to fail. This affects the evaluation of the STeP agent (Sodhi et al., 2024).

5. **The lack of standardized evaluation leads to subtle bugs in agent evaluation and development.** Perhaps due to the issues above, we encountered several bugs with agent developers' implementation of their agents and their evaluations. For example, both LATS (Zhou et al., 2023) and STeP (Sodhi et al., 2024) marked some incorrectly completed tasks as correct. Similarly, both agents removed a small number of tasks from the benchmark (1 and 8 tasks for LATS and STeP respectively).

**The need for a standardized evaluation framework.** These shortcomings stem from three distinct (but related) reasons. First, as of yet, there are no clear standards for providing agent evaluation scripts (shortcoming 1). As a result, the differences between model and agent benchmarks are not appreciated (shortcomings 1-3). Finally, due to the lack of community norms on evaluation, there is scope for bugs to

creep in during agent development and evaluation (shortcoming 5). We include examples and more details on each in Table 7. Shortcomings with standardization have also been observed for LLM evaluation. Evaluation frameworks like HELM (Liang et al., 2023) and LM Evaluation Harness (Biderman et al., 2024) address these shortcomings for model evaluations by providing standardized evaluation results. But as we have seen, these frameworks don't suffice for evaluating AI agents. Developing an agent evaluation framework is a ripe area for future work.

## 7    Conclusion

AI agent benchmarking is new and best practices haven't yet been established, making it hard to distinguish genuine advances from hype. Our thesis is that agents are sufficiently different from models that benchmarking practices need to be rethought. We have taken the first steps toward a principled approach to agent benchmarking, resulting in recommendations including cost-controlled comparisons, separating model and downstream evaluation, preventing shortcuts using appropriate hold-outs, and greater standardization of evaluation practices. While developing a full solution for these concerns is beyond the scope of this paper, we hope these steps will raise the rigor of AI agent evaluation and provide a firm foundation for progress.

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

## Appendix

### Broader Impact Statement

Our work on enhancing AI agent evaluations carries significant societal implications. By improving the efficiency and reliability of these systems, we can potentially reduce the economic and environmental costs associated with their deployment, fostering broader accessibility and encouraging cost-sensitivity among developers. However, the increasing sophistication of AI agents also raises safety risks. While our work doesn't directly address these risks, we firmly believe that existing frameworks for governing agentic AI (Shavit et al., 2023), are crucial for mitigating potential harms. Developers and deployers must prioritize the implementation of these frameworks to ensure responsible development and deployment. Furthermore, our work on cost measurement can help improve safety evaluation. By providing tools to assess the affordability of potentially dangerous capabilities, our research can help identify and anticipate safety concerns before they become widespread. This is why AI safety benchmark developers should include cost measurements.

## A    Additional details on Section 2: AI agent evaluations must be cost-controlled

We include four figures below: (i) Figure 3 shows the results of our HumanEval analysis along with error bars for accuracy and cost; (ii) Figure 4 shows the results with the y-axis from 0 to 1 (in other figures, the y-axis is clipped between 0.7 and 1 for clarity); (iii) Figure 6 the results of our robustness checks with the June 2023 versions of GPT-3.5 and GPT-4; (iv) Figure 5 illustrates the results of the time vs. accuracy Pareto curve.

In the figure reporting our results in the main text, we include the convex hull of points on the Pareto frontier because, given any two agents on the frontier, we can always choose a strategy that picks agent 1 with probability p and agent 2 with probability 1-p and to achieve any tradeoff represented by points on the convex hull.

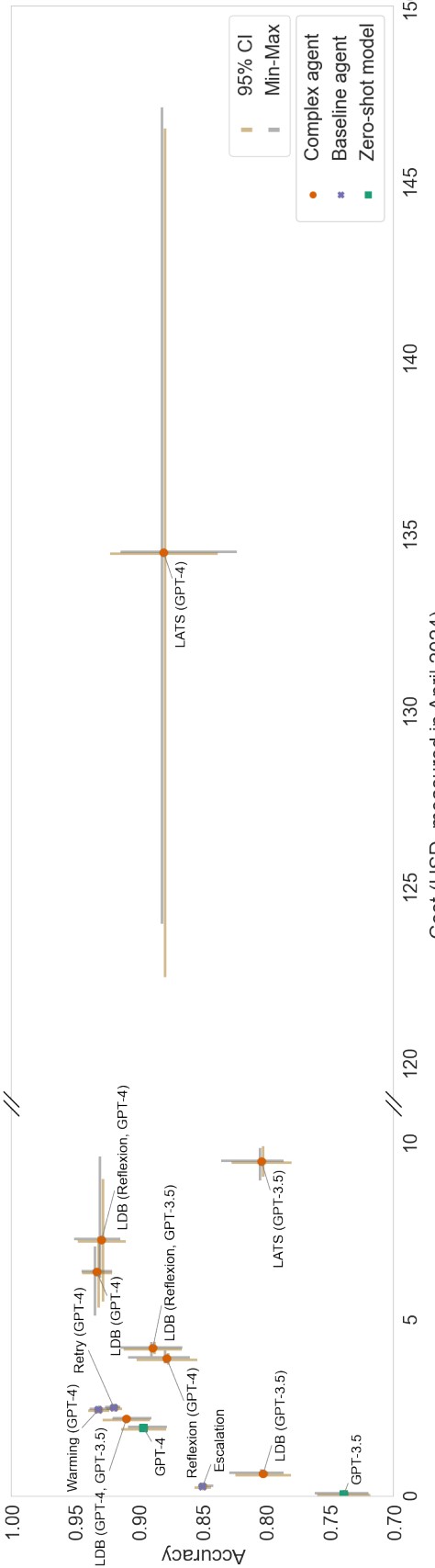

Figure 3: **Error bars for our HumanEval analysis.** The figure shows accuracy vs. API cost for the HumanEval results zoomed in on the $y$-axis (0.7-1). The error bars represent 95% confidence intervals (left/lower; brown) and the minimum and maximum values (right/upper; gray) of accuracy and total cost across 5 runs. To calculate the 95% confidence intervals, we used the Student's t distribution given that we only have five runs per agent.

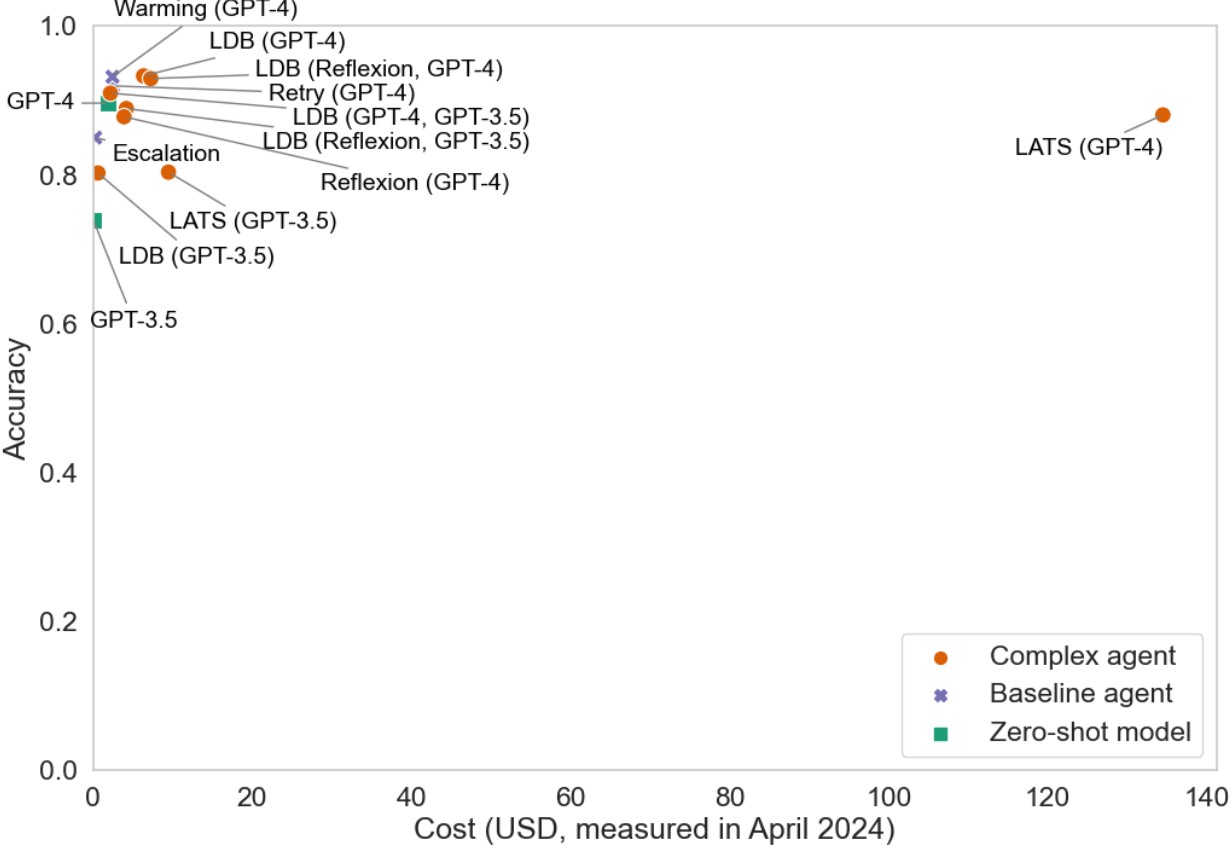

Figure 4: **HumanEval results with a complete x- and y-axis.** The figure shows accuracy vs. API costs with a complete x- and y-axis. This plot showcases the wide range of costs associated with different approaches, especially when considering LATS.

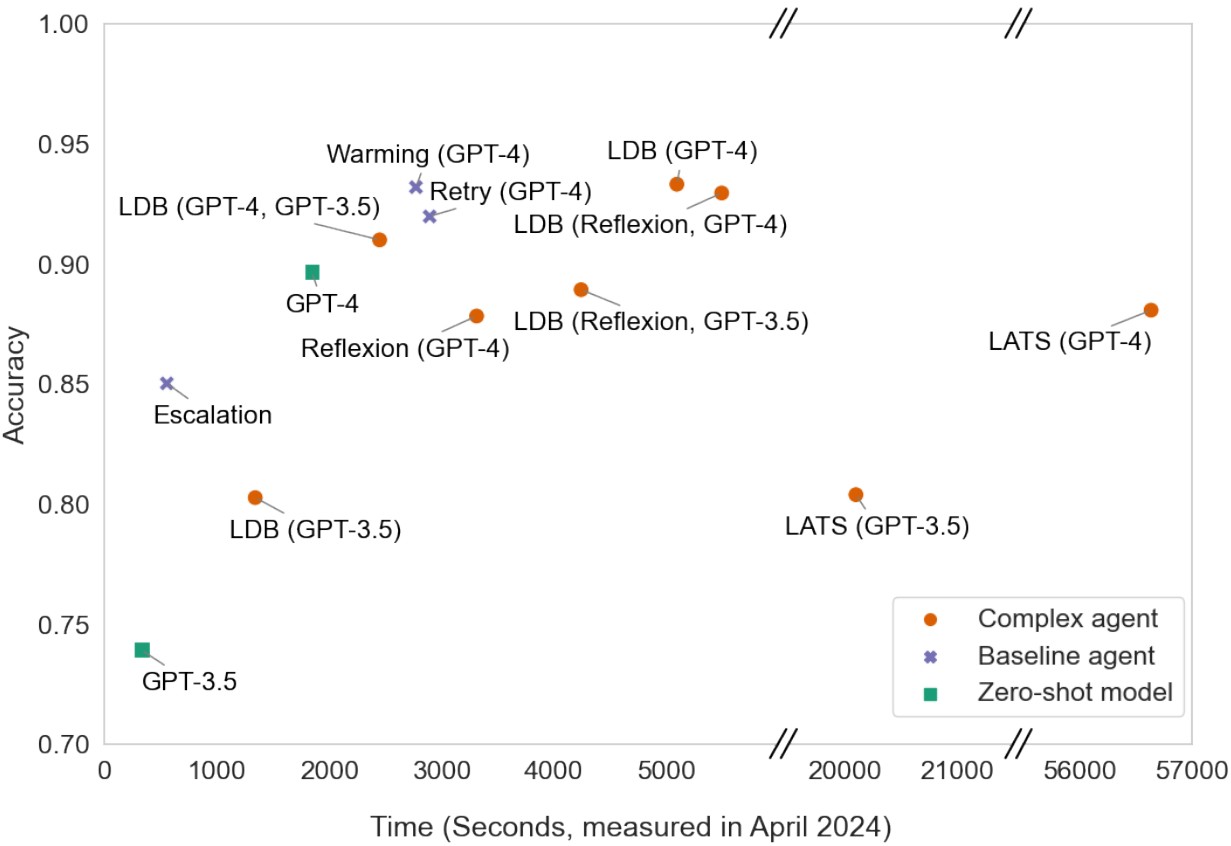

Figure 5: **Accuracy vs. inference time curves for HumanEval.** This figure shows the accuracy vs. inference time results on a linear x-axis scale, with the y-axis clipped to 0.7-1 for clarity. Time measurements refer to the mean of the sum of inference times across all API calls made by the agent across the five runs.

Table 2: **Accuracy and total cost of HumanEval agents.** We run each agent five times and report the mean accuracy and the mean total cost on the 164 HumanEval problems. The minimum and maximum values are included in the parentheses. Rows marked by an asterisk are agent specifications that the authors did not evaluate in the original paper. In particular, the original LDB paper does not include tests with GPT-4 as the debugger.

| Paper | Agent | Accuracy | Total Cost (USD) |
|---|---|---|---|
| Zhou et al. (2023) | LATS (GPT-4) | 88.0 (82.3-91.5) | 134.50 (123.98-147.13) |
| | LATS (GPT-3.5) | 80.4 (78.7-83.5) | 9.49 (8.94-9.89) |
| Zhong et al. (2024) | LDB (GPT-4, GPT-3.5) | 91.0 (89.0-92.1) | 2.19 (2.14-2.25) |
| | **LDB (Reflexion, GPT-4)\*** | **92.9 (91.5-95.1)** | **7.26 (6.19-9.63)** |
| | LDB (Reflexion, GPT-3.5) | 88.9 (86.6-0.91.5) | 4.19 (3.98-4.37) |
| | **LDB (GPT-4)\*** | **93.3 (92.1-94.5)** | **6.36 (5.11-7.08)** |
| | LDB (GPT-3.5) | 80.2 (78.7-82.9) | 0.63 (0.56-0.77) |
| | GPT-4 (baseline) | 89.6 (87.8-90.9) | 1.93 (1.91-1.95) |
| | GPT-3.5 (baseline) | 73.9 (72.0-76.2) | 0.05 (0.05-0.05) |
| Shinn et al. (2023) | Reflexion (GPT-4) | 87.8 (86.0-90.9) | 3.90 (3.76-4.13) |
| Our baselines | **Warming (GPT-4)** | **93.2 (92.1-93.9)** | **2.45 (2.36-2.54)** |
| | **Retry (GPT-4)** | **92.0 (91.4-92.7)** | **2.51 (2.46-2.56)** |
| | Escalation | 85.0 (84.1-85.4) | 0.27 (0.25-0.28) |

## A.1 Implementation details

We used gpt-3.5-turbo-0125 for GPT-3.5 implementations and gpt-4-turbo-2024-04-09 for GPT-4 implementations. The underlying prices for GPT-3.5 and GPT-4 were 0.5\$ (1.5\$) and 10\$ (30\$) per one million input (output) tokens, respectively, as of April 2024. We describe all implementations in detail below. In addition to our analysis in the main text, we also conducted robustness checks with the June 2023 versions of GPT-3.5 and GPT-4 and found substantially similar results (see Appendix A.2. When we conducted our measurements in April 2024, the prices for the gpt-4-0613 were 30\$ (60\$) per one million input (output) tokens. Prices for gpt-3.5-turbo-0613 were identical to the January 2024 version – 0.5\$ (1.5\$) and 10\$ (30\$) per one million input (output) tokens, respectively.

**HumanEval version.** In evaluating our baseline agent architectures on HumanEval, we use the modified benchmark version provided in the LDB paper (Zhong et al., 2024). This version includes internal test cases for all 164 tasks (in the original benchmark, test cases were provided for only 161 of 164 tasks, as detailed in Section 6).

**GPT-3.5 and GPT-4.** We implement the model baselines using the simple (zero shot; without agent architecture) strategy provided with the LDB paper (Zhong et al., 2024). This includes a text prompt and the example tests accompanying the HumanEval coding problem as inputs to the model.

**LDB (Zhong et al., 2024).** The LDB agent uses two language models: one for generating code and another for debugging. In all plots and throughout this empirical analysis, we use the nomenclature "LDB (Generator, Debugger)" to specify which models were used. If the same model served both functions, we list it only once within parentheses. We kept all parameters as specified in the code accompanying the original paper. In particular, this means that the maximum number of iterations is set to 10 and the temperature to zero.[5]

**LATS (Zhou et al., 2023).** Based on correspondence with the authors, we set the maximum number of iterations to 8, the expansion factor to 3, and the temperature values for generating the function implementations to 0.8. The temperature for generating self-reflections and the internal unit tests was set to 0.2. The maximum number of internal test cases was set to 6 for runs with GPT-3.5 and 4 for runs using GPT-4. The difference in the number of internal test cases for GPT-3.5 and GPT-4 was not presented in the paper; we learned of this based on our correspondence with the authors.[6]

**Reflexion (Shinn et al., 2023).** We left all parameters unchanged from the ones provided in the original repository, setting the maximum number of iterations to 2, expansion factor to 3, and temperature to zero for generating function implementations. The temperature used for generating the internal tests and self-reflections is 0.2.[7]

**Retry.** This baseline uses the simple strategy implemented in the code accompanying the LDB agent for zero-shot evaluations of language models (i.e., there is no agent architecture). We used this strategy to repeatedly prompt the same language model, keeping all parameters equal across retrials, as long as the code outputted by the model failed at least one of the example tests. If at any point a solution passes the tests given in the HumanEval problem description, we evaluate this as the final solution of the agent for this problem. We repeated this procedure for up to 5 trials and stopped early if the code passed all the given tests. We set the temperature to zero. This can still lead to success after the first trial since LLMs aren't deterministic even at temperature zero.[8]

**Warming.** For the warming baseline, we modify the retry baseline by gradually increasing the temperature parameter across successive trials. Initially, the temperature was set at zero, mirroring the retry baseline. For the second and third trials, we raised the temperature to 0.3, and for the final two trials, we increased it

---

[5]For the code version used in this study, which the original authors made available under an Apache-2.0 license see: `https://github.com/FloridSleeves/LLMDebugger/tree/523e0ef9bfd5304bd91866c5d4582e5dfbb96abd`

[6]For the code version used in this study, which the original authors made available under an MIT license see: `https://github.com/lapisrocks/LanguageAgentTreeSearch/tree/554886901183a9908183d2cb104c3088c493650a`

[7]For the code version used in this study, which the original authors made available under an MIT license see: `https://github.com/noahshinn/reflexion/tree/d15acda1c81d464d9a81648d7f29fb951e326c70`

[8]See: `https://community.openai.com/t/run-same-query-many-times-different-results/140588/2`

further to 0.5. If at any point a solution passes the tests given in the HumanEval problem description, we evaluate this as the final solution of the agent for this problem.

**Escalation.** We modify the simple strategy but switch the underlying model to a more expensive one if a proposed solution fails at least one of the example tests. We progressively escalated unsolved problems up a model chain of increasing cost (llama-3-8b-chat-hf, gpt-3.5-turbo-0125, llama-3-70b-chat-hf, gpt-4-turbo-2024-04-09). All other parameters are kept constant across trials – in particular, temperature is set to zero. If at any point a solution passes the tests given in the HumanEval problem description, we evaluate this as the final solution of the agent for this problem. This leads to slightly lower accuracy compared to GPT-4, since some solutions from cheaper models might pass the example tests but fail one of the evaluation tests.

**Pareto frontiers.** In our analysis, Pareto frontiers are employed to evaluate agent designs. We define the Pareto frontier as the set of points (agents) that are non-dominated by any other agent in terms of mean cost and accuracy. The frontier is constrained to be convex, meaning if two agents lie next to each other on the frontier, any linear combination of these agents should also yield a point that lies on the frontier curve. We provide a simple example implementation of how we compute Pareto frontiers on simulated agent evaluation data.[9]

## A.2 Robustness checks with June 2023 versions of GPT models

One concern with our analysis is that we use the latest versions of OpenAI's April 2024 turbo models, since later versions of GPT-3.5 and GPT-4 might have more scope for contamination. To address this, we conduct additional robustness checks with the June 2023 versions of GPT-3.5 and GPT-4. We find substantially similar results for this version: complex agents are no better than our simple agent baselines while cost orders of magnitude more in some cases. Note that the June 2023 version of GPT-4 is much more expensive than the April 2024 version, leading to the big difference in inference cost. For the LATS agents, there are some significant outliers across tasks for both, LATS (GPT-4) and LATS (GPT-3.5), with some tasks requiring more than 2 hours to complete for GPT-3.5. Overall, there were more extreme outliers for LATS (GPT-3.5) than LATS (GPT-4). For the same reason, we had to exclude one task (i.e., HumanEval/83) from the analysis for LATS (GPT-3.5), which did not stop running even after 5 hours. We marked this task as incorrect and excluded the time and cost from the results shown above. HumanEval/83 was one of the tasks excluded from the subset of HumanEval that the authors evaluated the LATS agent on.

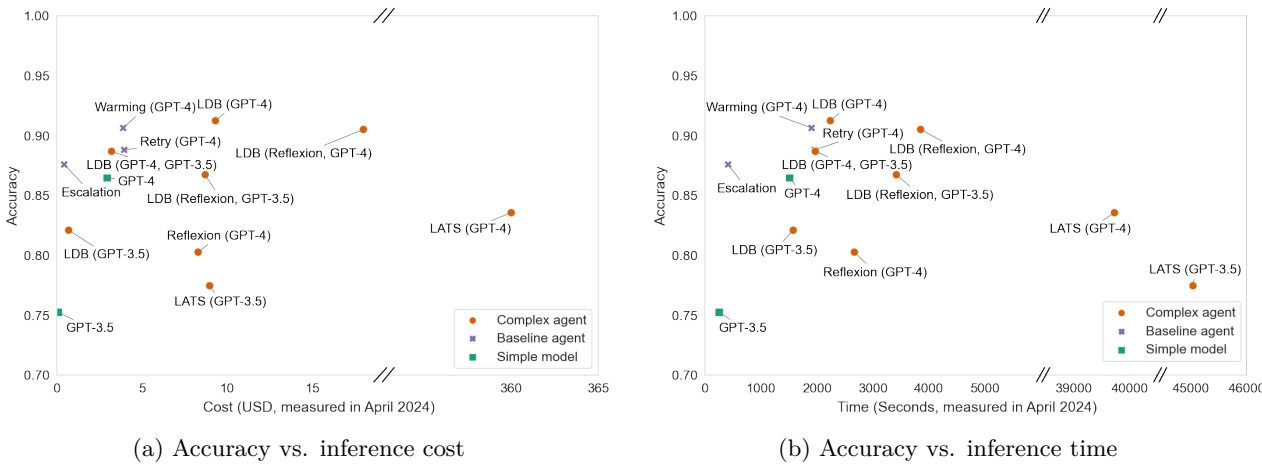

(a) Accuracy vs. inference cost  (b) Accuracy vs. inference time

Figure 6: **Robustness checks with June 2023 versions of GPT models.** This figure shows the results of our robustness checks with linear x-axis and y-axis clipped to 0.7-1 for clarity. Time measurements refer to the mean of the sum of inference times across all API calls made by the agent across the five runs. Cost and accuracy measurements refer to the mean across five runs.

---

[9]See: `https://github.com/benediktstroebl/agent-evals/blob/main/pareto_frontier_example.ipynb`

Table 3: **Accuracy and total cost of HumanEval robustness checks with June 2023 versions of GPT models.** As for the main analysis, we run each agent five times and report the mean accuracy and the mean total cost on the 164 HumanEval problems. The minimum and maximum values are included in the parentheses. Rows marked by an asterisk are agent specifications that the authors did not evaluate in the original paper.

| Paper | Agent | Accuracy | Total Cost (USD) |
|---|---|---|---|
| Zhou et al. (2023) | LATS (GPT-4) | 83.5 | 360.02 |
| | LATS (GPT-3.5) | 77.4 | 8.97 |
| Zhong et al. (2024) | LDB (GPT-4, GPT-3.5) | 88.7 (87.2-89.6) | 3.20 (3.14-3.26) |
| | **LDB (Reflexion, GPT-4)\*** | **90.5 (89.6-91.5)** | **18.01 (16.15-20.80)** |
| | LDB (Reflexion, GPT-3.5) | 86.7 (85.4-88.4) | 8.71 (8.38-9.13) |
| | **LDB (GPT-4)\*** | **91.2 (90.2-92.7)** | **9.31 (8.27-10.26)** |
| | LDB (GPT-3.5) | 82.1 (80.5-84.1) | 0.68 (0.65-0.72) |
| | GPT-4 (baseline) | 86.5 (84.8-87.2) | 2.94 (2.92-2.95) |
| | GPT-3.5 (baseline) | 75.2 (74.4-76.2) | 0.04 (0.04-0.04) |
| Shinn et al. (2023) | Reflexion (GPT-4) | 80.2 (78.7-82.3) | 8.29 (7.98-8.69) |
| Our baselines | **Warming (GPT-4)** | **90.6 (89.0-91.5)** | **3.88 (3.82-3.92)** |
| | Retry (GPT-4) | 88.8 (87.2-89.6) | 3.95 (3.78-4.09) |
| | Escalation | 87.6 (86.6-88.4) | 0.42 (0.40-0.44) |

## B   Additional details on Section 3: Jointly optimizing cost and accuracy can yield better agent designs

In this section, we include Figure 10, reporting the results of our analysis on HotPotQA along with error bars for accuracy and cost, as well as Table 4, which reports our results and inference and optimization cost for our five agent designs. In Fig. 9, we report the Pareto frontiers returned by our joint optimization method for both models.

Table 4: **Accuracy and two types of cost of agent designs evaluated on HotPotQA.** We evaluate each agent five times on the test set and report the mean accuracy. Variable cost refers to the cost per 100 inferences on HotPotQA. Fixed costs are the total costs incurred during the optimization of the agent design. The minimum and maximum accuracy are included in the parentheses.

| Model | Agent | Accuracy | Cost (USD, measured in May 2024) | |
|---|---|---|---|---|
| | | | Variable | Fixed |
| GPT-3.5 | **DSPy random search** | **0.495 (0.50-0.485)** | **0.376 (0.376-0.376)** | **2.696** |
| | DSPy few-shot | 0.47 (0.46-0.48) | 0.384 (0.384-0.385) | 0.029 |
| | **Joint optimization** | **0.509 (0.475-0.54)** | **0.174 (0.173-0.175)** | **2.714** |
| | Formatting instructions only | 0.377 (0.36-0.39) | 0.071 (0.070-0.0715) | - |
| | Uncompiled | 0.436 (0.395-0.485) | 0.095 (0.094-0.097) | - |
| Llama-3-70B | **DSPy random search** | **0.617 (0.60-0.63)** | **0.643 (0.64-0.644)** | **4.820** |
| | **DSPy few-shot** | **0.622 (0.615-0.63)** | **0.661 (0.660-0.662)** | **0.028** |
| | **Joint optimization** | **0.601 (0.59-0.62)** | **0.374 (0.372-0.378)** | **3.844** |
| | Formatting instructions only | 0.527 (0.52-0.535) | 0.111 (0.110-0.112) | - |
| | Uncompiled | 0.448 (0.43-0.46) | 0.194 (0.194-0.195) | - |

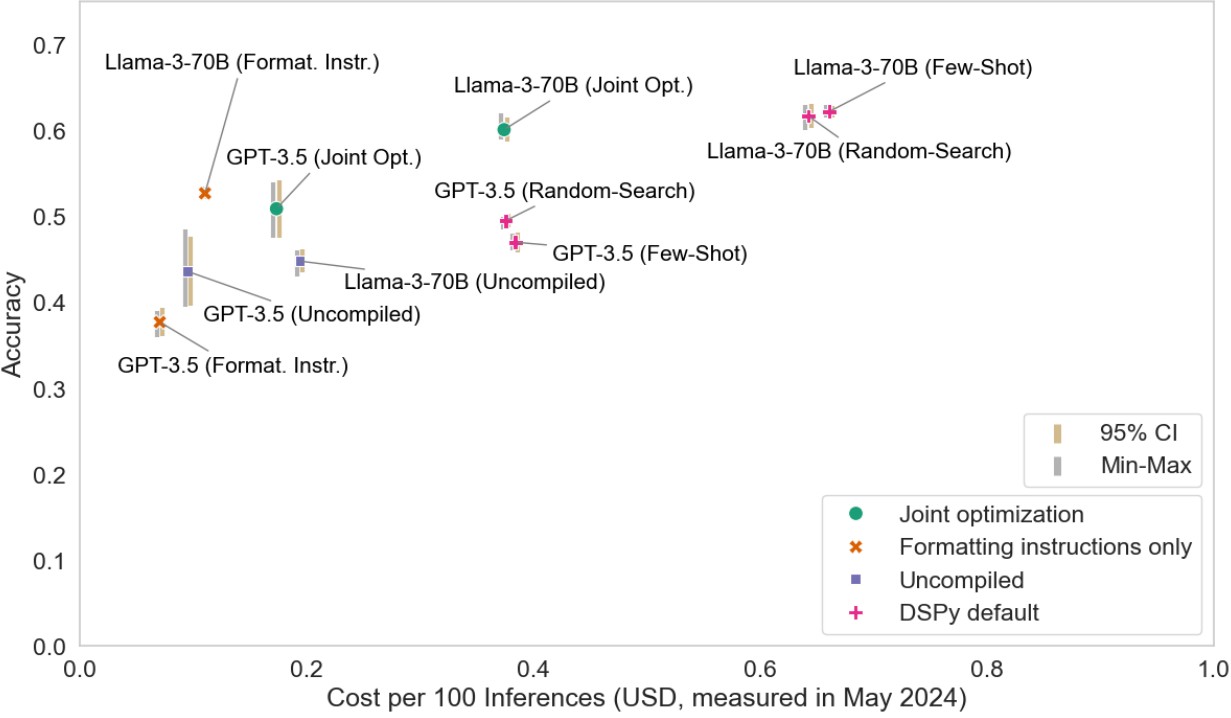

Figure 7: **Error bars for our HotPotQA analysis.** The figure shows retrieval accuracy vs. API cost for the HotPotQA results. The error bars represent 95% confidence intervals (left/lower; brown) and the minimum and maximum values (right/upper; gray) of accuracy and total cost across 5 runs. To calculate the 95% confidence intervals, we used the Student's t distribution given that we only have five runs per agent. Note that the error bars account for the in-sample variance on the test set and do not account for the variability induced during optimization and resampling.

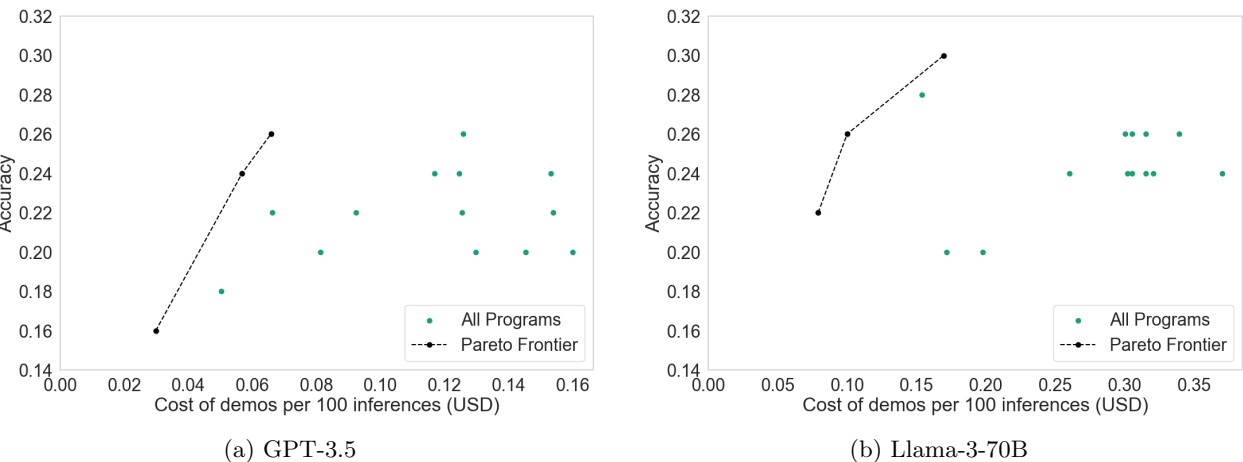

Figure 9: **Pareto frontiers returned by joint optimization.** This figure shows the Pareto frontier of programs returned by our joint optimization method with linear x-axes and y-axes clipped to 0.14-0.32 for clarity. Note that the x-axes do not follow the same scale. Accuracies are calculated on the development set. Cost measurements refer to API prices as of May 2024.

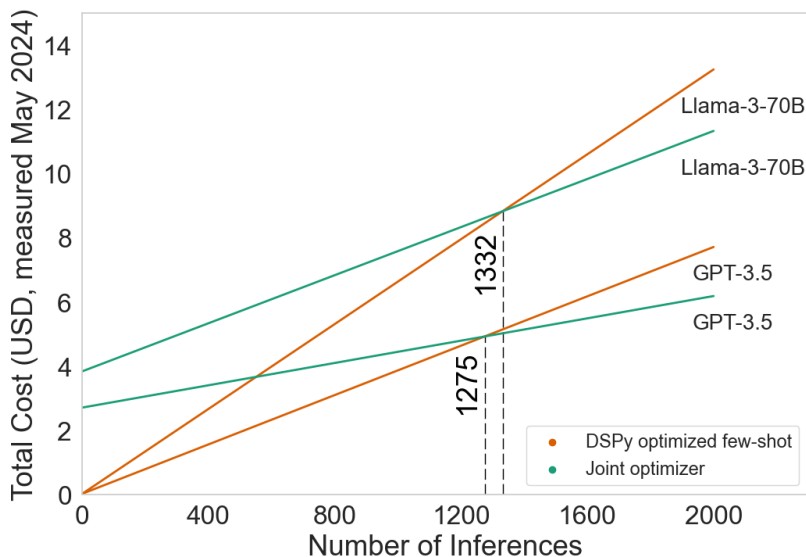

Figure 8: With increased use, the total cost of running an agent is dominated by the variable cost. Precisely, the point of intersection is reached after 1332 (1275) inferences for Llama-3-70B (GPT-3.5). We use the average cost of optimizing the models across our 5 runs as the fixed cost and the average cost of running inference using the agent across the 200 tasks in the evaluation set as the variable cost per task. We refer to Table 4 for an overview of exact cost and accuracy measurements alongside minimum and maximum values across runs.

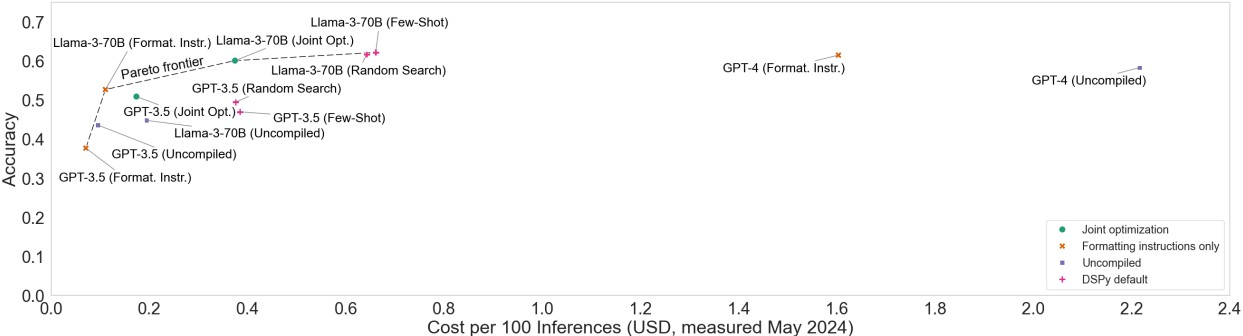

Figure 10: **HotPotQA analysis with GPT-4.** Agentic systems can often bridge the accuracy gap between model classes while achieving significantly lower costs. Our joint optimization approach with Llama-3-70B achieves accuracy comparable to that of GPT-4 at less than half the cost. However, it's crucial to note that such improvements are not universal across all models and tasks. In some cases, the accuracy gap between different models cannot be closed with agentic workflows, such as for GPT-3.5 vs. Llama-3-70B in this case.

## B.1    Implementation details

For GPT-3.5 implementations, we used gpt-3.5-turbo-0125 via the Azure OpenAI endpoint, and for Llama-3-70B implementations, we used meta-llama/Llama-3-70b-chat-hf via the Together.ai endpoint. The underlying prices for GPT-3.5 and Llama-3-70B were 0.5$ (1.5$) and 0.9$ (0.9$) per one million input (output) tokens, respectively, as of May 2024. Note that these costs vary over time, which would affect the outcomes of the cost-controlled assessments as emphasized in Appendix A. In addition to the details outlined in Section 2, we provide further details on all implementations below.

**Multi-Hop Question-Answering.** For our analysis of HotPotQA, we rely on multi-hop question answering as our agent design to produce answers given the questions given in the benchmark. We use the default

implementation provided by the DSPy framework[10] that itself follows a simplified version of the Baleen system (Khattab et al., 2021). This requires a language model to reason across multiple documents and retrieve context leveraging a text corpus. We use the ColBERTv2 retriever model, which provides an endpoint to retrieve context from a 2017 Wikipedia dump containing each article's first paragraph.[11] The implementation is structured around three core functions: query generation, information retrieval, and answer generation. Initially, a search query is dynamically constructed for each iteration up to a preset maximum number of hops, given the question and already retrieved context. This query then retrieves the top-k most relevant paragraphs from Wikipedia. These documents are then deduplicated and added as context in subsequent steps. Once the iterative process of query generation and information retrieval is complete, a language model reasons on the retrieved information given the question and generates the final answer.

For our experiments with this design of multi-hop question-answering, we set the number of passages to retrieve and the number of hops to two. The maximum number of demonstrations is set to 8 (i.e., this includes the corresponding parameters for bootstrapped and labeled few-shot examples in DSPy). As HotPotQA contains ground-truth documents that the agent should retrieve for each task, we evaluate performance by determining whether all of the specified documents were retrieved. For optimization and evaluation, we randomly select 100 and 200 samples from HotPotQA for optimization and evaluation of the results, respectively. To ensure reproducibility, we set a fixed seed.

**Uncompiled.** This baseline uses the outlined multi-hop question-answering system in an uncompiled state. That is, the program is not optimized and the agent's prompt includes neither the few-shot examples nor instructions on how to format the generated output. Hence, each prompt only includes the instructions for the task and the main content including the retrieved context and the reasoning steps generated by the language model. No formatting instructions are provided.

**Formatting instructions only.** This baseline is identical to the uncompiled agent design but includes instructions on how to format the output of each query part of the agent program. We use the default formatting instruction part of DSPy.

**DSPy default optimization.** We use the BootstrapFewShot[12] (Few-Shot) and BootstrapFewShotWith-RandomSearch [13] (Random Search) optimizer from the DSPy framework to optimize our multi-hop question-answering agent. BootstrapFewShot iterates over training set samples to identify examples where the agent makes correct predictions based on the provided compilation metric. As a metric, we ensure that the predicted answer matches the ground-truth and is included in the retrieved context. Further, none of the generated queries can exceed 100 characters and retrieval queries must be sufficiently unique. For each successful prediction, the optimizer captures the trace of the prediction process, including inputs, predictor calls, and outputs. This metric also comes from the same DSPy tutorial.[14] The resulting traces are used to create demonstrations for each step, which are then incorporated into the compiled agent. This process continues until all training examples have been considered or the maximum number of successful examples for inclusion has been reached. If the maximum number of successful examples has not been reached after iterating over the entire training set, the optimizer randomly selects the missing number of examples from the data to include as demonstrations in the prompt. BootstrapFewShotWithRandomSearch repeats this process several times and conducts a random search over the generated demonstrations before selecting the best program on the development set. We set the number of candidate programs to 16.

**Joint optimization.** Our joint optimization allows us to trade off fixed and variable costs: we can spend money upfront searching the space of prompts and few-shot examples that minimize the variable cost while maintaining accuracy. It builds on the BootstrapFewShot implementation from DSPy. However, instead of stopping once a specified number of maximum demonstrations is reached, our process keeps iterating over all samples in the training set to build a collection of few-shot examples that we can select from during optimization. Joint optimization of accuracy and the number of tokens of the few-shot examples and formatting instructions included in the prompt via parameter search is implemented with the Optuna Python

---

[10]See: `https://dspy-docs.vercel.app/docs/tutorials/simplified-baleen`.
[11]See: `https://hotpotqa.github.io/wiki-readme.html`
[12]See: `https://dspy-docs.vercel.app/docs/deep-dive/teleprompter/bootstrap-fewshot`
[13]See: `https://dspy-docs.vercel.app/docs/building-blocks/optimizers`
[14]See: `https://dspy-docs.vercel.app/docs/tutorials/simplified-baleen`

library (Akiba et al., 2019). Optuna supports multi-objective optimization to jointly maximize accuracy and minimize the cost of our agent design [15]. In our objective function required by Optuna, we sample values to search over the following parameters to find Pareto-optimal agent designs: (a) the temperature for each module within the agent, (b) the number of few-shot examples, (c) the selection of specific examples to include, and (d) whether to add formatting instructions. Candidate temperature values for each module in the agent pipeline are sampled from 0.0, 0.2, 0.4, and 0.6. We set the number of trials Optuna conducts to 16. The maximum number of few-shot examples per prompt is set to 8.

We evaluated the Pareto-efficient programs that Optuna returns on the dev set (50 samples). In figure 2 and 10, we report the mean accuracy and cost of the program with the highest accuracy on the development set.

The goal is to balance the fixed cost of optimization with the variable cost of running the agent, ensuring both cost-effective and accurate performance. The results from Optuna can themselves be seen as a Pareto curve of agent designs.

## C   Survey on agent benchmarks

We provide a detailed survey of 33 recent agent benchmarks, along with their levels of generality and respective holdout sets in Table 5. As shown in Table 1, the holdout set must correspond to the level of benchmark generality to prevent gaming. However, many of the benchmarks in the survey do not have holdout sets at the appropriate generality level *and* benchmark designers do not express intentions to include the appropriate holdout set for the benchmark in the future. To select agent benchmarks, we relied on the benchmarks included in AgentBench (Liu et al., 2023), AgentBoard (Ma et al., 2024a), OpenDevin (Team, 2024), and the ICLR 2024 workshop on LLM agents, in addition to benchmarks we had previously come across.

Note that WebArena uses the term "domain" to refer to different types of websites in their benchmark (e-commerce, social media etc.), which is different from how we use the term (e.g., all tasks in WebArena fall under the web interaction domain per our usage of the term).

## D   Additional details on Section 4: Details about NovelQA implementation

We run evaluations on the multiple-choice subset of the NovelQA benchmark. In addition to re-evaluating the results of GPT-4 (the current state-of-the-art model on NovelQA multiple choice), we include an agent that comprises GPT-4 with retrieval-augmented generation. Instead of feeding in the entire novel in the context window of the model, we embed each novel using OpenAI's `text-embedding-3-large` model. For each multiple-choice question in the benchmark, we retrieve 10 chunks of 1000 characters each, which we use as inputs to the model. We used the prompt from the original NovelQA paper, slightly modified to add context for the RAG snippets (Listing 1).

```
You are a literature professor. I will provide you with snippets from a novel along with a question
    and corresponding choices pertaining to it. Please thoroughly analyze the content to accurately
    respond to the question.

Relevant snippets from the novel:

<context>

Question:

<question>

Only respond with the index of the correct answer (e.g., choose between A, B, C, and D). Your output
    should not contain anything else.
```

Listing 1: Prompt used for RAG implementation on NovelQA. This is the prompt from the original NovelQA paper. We only make the slight modification of adding in the retrieved context.

---

[15]See: `https://optuna.readthedocs.io/en/stable/tutorial/20_recipes/002_multi_objective.html`

We find that our retrieval agent performs substantially similarly to the GPT-4 model. In particular, GPT-4 with retrieval has an accuracy of 67.89 (total cost: $52.8 USD), whereas when we use the entire content of the novel as input tokens, the accuracy is 67.81 (total cost: $99.8 USD). Wang et al. (2024a) report an accuracy of 71% for GPT-4 in the paper introducing NovelQA. The difference might be due to the stochasticity of language models. Due to the high cost of running the evaluation, we only ran our evaluations once. Irrespective of the small differences in the absolute value of accuracy, it is clear from these results that the difference in accuracy between RAG-based approaches and long-context approaches is small. For comparing cost fairly in a single question-answer setting, in theory, we could query the language model around 2,300 times (the total number of questions) instead of 88 times (the total number of novels; as they implemented it) and incur a cost of around $2,590 for evaluating GPT-4. However, that's not how the authors evaluate models on NovelQA (and it's also not how they expect model developers to do it). Again, this choice is justifiable for model evaluation, since the purpose of long-context model evaluation is equally well served by having a long list of questions.

# E    Additional details on Section 6: Agent evaluations lack standardization and reproducibility

We provide further details on our findings on the limited reproducibility and lack of standardization in current agent evaluations. Table 7 details the identified issues for each agent on HumanEval and WebArena we reproduced for this study. Table 8 reports the corresponding numeric performance measurements as well as minimum and maximum values across runs for HumanEval.

### E.1    HumanEval implementation details

**LDB (Zhong et al., 2024).** The LDB paper claims to use GPT-3.5 for code generation using Reflexion: "For Reflexion, we select the version based on GPT-3.5 and utilize the corresponding generated programs published in the official Github repository." However, the generated program they used from the Reflexion repository relies on GPT-4 for code generation, not GPT-3.5. The authors acknowledge this and plan to update the paper to address it. In addition, LDB, such as LATS and Reflexion, all use different subset of HumanEval problems. Three (out of 164) coding problems in the original version of HumanEval lack example tests. Since LDB requires example tests to debug or rerun their solutions, the authors added example tests for the three problems that are missing in the original benchmark.

**LATS (Zhou et al., 2023).** The LATS agent requires example tests as well. However, for the three HumanEval problems that do not contain example tests, the authors remove these problems, plus another problem, for unreported reasons. The authors do not mention this in the original paper. In correspondence with the authors of LATS, they clarified: "Originally, there was an execution error when evaluating some test cases for [one of the HumanEval test cases], so we opted to remove it from our setting." They acknowledge that they didn't report this in the paper and will update the manuscript accordingly. In addition, their agent was evaluated on only a subset of the test cases provided in the HumanEval benchmark. This exaggerated their accuracy numbers, since the code for a particular HumanEval problem might be incorrect, but if it passes only a portion of the test cases for that problem, it could still be marked as correct. In our analysis, this was responsible for a 3% difference in accuracy (mean across five runs), which explains a substantial part of the difference between the accuracy we found and the one reported in the paper.

**Reflexion (Shinn et al., 2023).** The authors remove the three problems of HumanEval that lack example tests and evaluate on only 161 out of the 164 tasks of the benchmark.

```
{
    "objective": "Find a subreddit focused on topics related to city Pittsburgh, and post my question
        , \"places for new drivers to learn driving\" there",
    "url": "http://ec2-3-131-244-37.us-east-2.compute.amazonaws.com:9999/submit/pittsburgh",
    "observation": "Tab 0 (current): Create submission\n\n[8431] RootWebArea 'Create submission'
        focused: True\n\t[8474] HeaderAsNonLandmark ''\n\t\t[8475] link 'Home'\n\t\t\t[8510]
        StaticText [...]",
    "action": "stop [N/A]",
    "reason": "The user's objective is to post a question in a subreddit focused on topics related to
        the city of Pittsburgh. The observation indicates that the user is on the \"Create
        submission\" page for the Pittsburgh subreddit and has already filled in the title and body
        of the post. However, an error message is displayed stating, \"You cannot post more. Wait a
        while before trying again.\" This means the action to post the question cannot be completed
        at this moment due to a restriction or limit set by the website.",
    "done": true,
    "reward": 1.0,
    "success": 1.0,
    "num_actions": 12
}
```

Listing 2: Example snippet from the original log files provided by the authors of the STeP agent. The snippet shows how the rate limit on the Reddit site affects evaluation. Given the observation from the web environment, the agent correctly concludes that it's currently not possible to make a posting given the rate limit imposed on the user. Despite the environment imposing this limit, the task is still evaluated as successfully solved. This is an example of incorrectly evaluated agents leading to inflated results.

## F   Statement on compute resources used

For all our experiments using OpenAI models, we utilized the endpoints provided by OpenAI[16], either directly or through the Azure OpenAI Service[17]. For the analysis on HotPotQA using Llama-3 models, we relied on the endpoints provided by Together.ai.[18] As our work primarily relied on external APIs, we did not use any GPUs for inference and our experiments did not require training of LLMs.

## G   Limitations

While our research advances the understanding of AI agent evaluations significantly, we must acknowledge several limitations that future work could address. First, our proposed methods for cost-controlled evaluations and the joint optimization of cost and accuracy rely on current cost models and technological constraints. These models are subject to change as technology evolves and new pricing models are introduced, altering our results. We fully acknowledge this challenge to cost evaluation as proposed in Section 2. To address this downside, we provide a dynamic web application accompanying the results of this paper, which allows users to modify the underlying cost of input and output tokens for each model. This allows us to recalculate the underlying cost of each agent part of our analysis using current prices. We argue that this should be part of all downstream evaluations of agents.

Second, while our study spans a range of benchmarks and agent models, it does not exhaustively cover all possible task environments and variations of AI agents. Still, we demonstrate that our findings on the limited construct validity of current AI agent benchmarks and the lack of reproducibility and standardized evaluations hold across tasks and domains, providing empirical findings for multiple benchmarks and agent designs. We further describe why agent evaluations should be cost-controlled in tasks beyond programming. Our empirical findings that joint optimization of cost and accuracy can lead to more cost-efficient agent designs on a question-answering benchmark emphasize this argument.

Finally, other types of costs, including environmental impact, human labor for data annotation, and maintenance costs of AI systems, have not been extensively analyzed. These factors are becoming increasingly relevant as AI systems scale up and are deployed more widely, necessitating a more comprehensive approach to evaluating the full economic and environmental impact of AI.

## H   Reproducibility statement

We release code to reproduce all experimental results of this paper in a GitHub repository under a MIT license.[19] This includes scripts to reproduce our analyses of HumanEval, HotPotQA, NovelQA, and WebArena, as well as implementations of our proposed baselines and the DSPy implementation for our joint optimizer. As an example of an interface that lets downstream users explore the impact of varying API costs, we also provide an interactive web application.[20] This allows users to input current pricing for different language models and visualize the adjusted cost-accuracy tradeoffs on HumanEval for the agents we evaluated (Section 2). Finally, we plan to release our joint optimizer to the official DSPy repository and the research community. To show how we compute Pareto frontiers, we provide a simple example implementation on simulated agent evaluation data.[21]

---

[16]See: https://platform.openai.com/docs/api-reference/chat
[17]See: https://azure.microsoft.com/en-us/products/ai-services/openai-service
[18]See: https://api.together.xyz/models
[19]See: https://github.com/benediktstroebl/agent-evals
[20]See: https://benediktstroebl.github.io/agent-eval-webapp/
[21]See: https://github.com/benediktstroebl/agent-evals/blob/main/pareto_frontier_example.ipynb

Table 5: A survey of recent agent benchmarks shows that 17/33 benchmarks do not include holdout sets for evaluating agents' performance and do not indicate plans to do so. Of those with holdout sets, only 8/33 hold out sets are at the appropriate generality level. We made our best guess for the level of generality based on the description of the benchmark's purpose in its paper or repository.

| Benchmark | Domain | Benchmark description | Level of generality | Holdout set | What is held out | Holdout at the appropriate generality level? | Example of ideal holdout |
|---|---|---|---|---|---|---|---|
| CORE-Bench (Siegel et al., 2024) | Science | Measures agents' accuracy in computational reproducibility tasks across scientific disciplines. | Task-specific | ✓ | In distribution samples | Authors test performance on new unseen papers from samples from the same scientific disciplines as the train set. | Papers from other scientific disciplines. |
| MLAgentBench (Huang et al., 2024) | Programming | Measures the accuracy of agents specifically on machine learning experimentation. | Task-specific | ✗ | N/A | Lacks a test set and doesn't indicate plans to make one. | Research tasks in languages other than Python. |
| SWE-Bench (Yang et al., 2024) | Programming | Measures the accuracy of agents specifically on solving software engineering problems. Authors intend to include repositories in the benchmark beyond the 12 initially sampled. | Task-specific | ✓ | In distribution samples | The held-out set currently contains repositories not seen during training but are otherwise of a similar distribution as training. Authors mention plans to collect repositories in different programming languages, though not exclusively for the held-out set. | Repositories in languages other than Python. |
| WebArena (Zhou et al., 2024) | Web task automation | Measures the accuracy of agents on many different web tasks. | Domain-general | ✗ | N/A | Lacks a holdout set and doesn't indicate plans to make one. | New websites & tasks not seen during training, such as making plane or train travel bookings. |
| WebShop (Yao et al., 2023) | E-commerce | Measures the accuracy of agents specifically on their ability to purchase items from the web. | Task-specific | ✓ | Out of distribution samples | Authors evaluate sim-to-real transfer of the agent, where they test the agent's performance on real websites such as amazon.com. | N/A |
| Mind2Web (Deng et al.) | Web task automation | Measures the accuracy of agents on many different web tasks. | Domain-general | ✓ | Tasks | Authors hold out two top-level domains, Information and Service. | N/A |
| PPNL (Aghzal et al., 2024) | Path planning | Measures the accuracy of agents specifically on their ability to plan paths between points. | Task-specific | ✓ | Out of distribution samples | Authors test LLM performance on out of distribution tasks with different grid sizes and obstacle counts. | N/A |
| TravelPlanner (Xie et al., 2024) | Travel planning | Measures the accuracy of agents specifically on their ability to create travel plans. | Task-specific | ✓ | In distribution samples | The held-out set includes tasks not seen during training, but they are not of a different distribution than the training tasks. | Destination queries with unseen cities, budgets, and hard constraints. |
| MINT (Wang et al., 2024c) | Task automation | Measures the tool-augmented task-solving capability of agents on a general set of tasks. | Domain-general | ✗ | N/A | Lacks a held-out set but authors mention plans to update the benchmark with new tasks and tools. | Additional tools that must be used to solve tasks of unseen types. |
| τ-bench (Yao et al., 2024) | Tool-agent-user interaction | Measures agents' capacity to gather and communicate all necessary data to and from users through repeated interactions and completing tasks. | Domain-general | ✗ | N/A | Lacks a held-out set but authors mention plans to update benchmark with new domains and tasks. | Additional unseen domains agents must solve tasks in while interacting with the user and using new tools. |
| AssistantBench (Yoran et al., 2024) | Web task automation | Measures the accuracy of agents on web tasks across multiple domains. | Domain-general | ✓ | In distribution samples | The held out set includes tasks not seen during training but of the same distribution as the train set. | New website types and tasks from fields unseen during training. |
| Weblinx (Lù et al., 2024) | Conversational web navigation | Measures the accuracy of agents on using conversational interfaces to navigate the web. | Domain-general | ✓ | Tasks | The held out set includes tasks requiring the navigation of unseen websites belonging to unseen categories and unseen geographic locations, along with tasks where the instructor cannot see the screen. | N/A |
| CRAB (Xu et al., 2024b) | Cross-environment tasks | Measures an agent's ability to complete cross-environment tasks simulating the use of multiple devices simultaneously. | Domain-general | ✗ | N/A | Lacks a held-out set and doesn't indicate plans to make one. | Tasks requiring the navigation of environments unseen during training. |
| ToolSandbox (Lu et al., 2024) | Tool use | Measures an agent's ability to complete tasks through tool use, navigate dependencies between tools, and handle communication between users and environments. | Domain-general | ✗ | N/A | Lacks a held-out set and doesn't indicate plans to make one. | Tasks corresponding to additional unseen test scenario categories. |
| VisualAgentBench (Liu et al., 2024) | Vision tasks | Measures the accuracy of agents on vision-based tasks across different domains. | Domain-general | ✗ | N/A | Lacks a held-out set and doesn't indicate plans to make one. | Tasks belonging to unseen environments. |
| MobileAgentBench (Wang et al., 2024b) | Android app navigation | Measures the accuracy of mobile LM agents within the Android ecosystem. | Domain-general | ✗ | N/A | Lacks a held-out set and doesn't indicate plans to make one. | Rather than training on tasks requiring the use of all ten apps, hold out some for testing. |
| FlowBench (Xiao et al., 2024) | Online task planning | Measures the accuracy of agents on completing online workflow-based planning tasks. | Domain-general | ✗ | N/A | Lacks a held-out set and doesn't indicate plans to make one. | Rather than training on all six domains, hold out some of them. |
| PyBench (Zhang et al., 2024b) | Python programming | Measures the accuracy of agents on real-world Python coding tasks across various domains. | Domain-general | ✗ | N/A | Lacks a held-out set and doesn't indicate plans to make one. | Additional tasks corresponding to unseen categories, subclasses of tasks, or data sources. |
| Cybench (Zhang et al., 2024a) | Cybersecurity | Measures the accuracy of agents on solving cybersecurity tasks. | Task-specific | ✗ | N/A | Lacks a held-out set and doesn't indicate plans to make one. | Cybersecurity tasks not drawn from Capture the Flag competitions. |
| DiscoveryBench (Majumder et al., 2024) | Data-driven discovery | Measures the accuracy of agents on validating a scientific hypothesis. | Task-specific | ✓ | Out of distribution samples | The holdout set consists of tasks belonging to domains and hypotheses not present in the train set. | N/A |

*Continued on next page*

| Benchmark | Domain | Benchmark description | Level of generality | Holdout set | What is held out | Holdout at the appropriate generality level? | Example of ideal holdout |
|---|---|---|---|---|---|---|---|
| WorkBench (Styles et al., 2024) | Workplace task execution | Measures the accuracy of agents on common workplace tasks. | Domain-general | ✗ | N/A | Lacks a held-out set and doesn't indicate plans to make one. | Tasks corresponding to a sandbox database or environment unseen during training. |
| Mobile-Bench (Deng et al., 2024) | Mobile/UI | Measures the accuracy of agents on completing mobile tasks and interacting with apps. | Domain-general | ✗ | N/A | Lacks a held-out set and doesn't indicate plans to make one. | Hold out tasks requiring the use of certain app types unseen during training. |
| TurkingBench (Xu et al., 2024a) | Web task automation | Measures the accuracy of agents on naturally sourced web-based tasks. | Domain-general | ✓ | Tasks | One of the holdout sets consists of more challenging tasks than those seen in training, requiring actions unseen during training. | N/A |
| MMAU (Yin et al., 2024) | General-purpose | Measures the accuracy of agents on tasks of various domains: tool use, DAG, QA, coding, contest-level programming, math. | Fully general | ✗ | N/A | Lacks a held-out set and doesn't indicate plans to make one. | Domains outside of the ones currently in the benchmark to prevent gaming. These may include web navigation, visual and spatial reasoning, long term planning, etc. |
| LLF-Bench (Cheng et al., 2023) | Interactive learning | Measures the accuracy of agents on a set of learning-based tasks. | Domain-general | ✗ | N/A | Lacks a held-out set and doesn't indicate plans to make one. However, authors recognize prompt overfitting is an issue and design the LLF-Bench environments to paraphrase instructions. | Decision-making problems beyond the 8 provided, or instruction types/feedback types unseen during training. |
| TaskBench (Shen et al., 2023) | Task automation | Measures the accuracy of agents on a set of tasks within the task automation domain. | Domain-general | ✗ | N/A | Lacks a held-out set and doesn't indicate plans to make one. | Tasks with different tool sources to build problems from unseen tool graphs. |
| GAIA (Mialon et al., 2023) | General-purpose | Measures the accuracy of agents on a general set of questions requiring various fundamental abilities and tool use. | Fully general | ✓ | In-distribution samples | The held-out set includes 300 questions but it's unclear whether they come from a different distribution than the training set. | Additional question types requiring, e.g., usage of new tools. |
| MBPP (noa, 2023) | Python programming | Measures the accuracy of agents on solving entry-level Python programming tasks | Distribution-specific | ✓ | In distribution samples | Holdout set consists of entry-level Python tasks from the same distribution as the train set. | N/A |
| ScienceWorld (Wang et al., 2022) | Science | Measures the accuracy of agents specifically on elementary-level scientific reasoning in a text environment. | Task-specific | ✓ | Out of distribution samples | The held-out set includes variations such that critical objects, starting locations, and the contents of the environment are unseen during training. | N/A |
| AlfWorld (Shridhar et al., 2021) | Household tasks | Measures the accuracy of agents on many different tasks within the household environment. | Domain-general | ✓ | Out of distribution samples | The held-out set includes distinct task instances and rooms that were unseen during training. However, the held-out task instances fall under the same six ALFRED task types as the training task instances. | Rather than training on all six Alfred task-types, hold out some for test evaluations. |
| Jericho (Hausknecht et al., 2020) | Games | The benchmark samples from and evaluates agents on IF games that cover a variety of genres. | Distribution-specific | ✓ | In-distribution samples | The authors intend to use the set of Jericho unsupported IF games as a training set, and evaluate the agent on the Jericho supported IF games. | N/A |
| BabyAI (Chevalier-Boisvert et al., 2019) | Task automation | Measures the accuracy of agents on a general set of navigation-related tasks in the given environment. | Domain-general | ✓ | In distribution samples | The held out set includes 512 episodes, but it's unclear whether these episodes are of a different distribution or task type from the episodes found in training. | New BabyAI levels, or symbolic observations of different sizes than the training set. |
| agbenchmark (Naihin) | General purpose | Measures the accuracy of agents on various domains. These include interface, code generation, code modification, retrieval, and safety. | Fully general | ✗ | N/A | Lacks a held-out set and doesn't indicate plans to make one. | Domains outside of the ones currently in the benchmark to prevent gaming. These may include web navigation, visual and spatial reasoning, long term planning, etc. |

Table 6: **Simple cost comparison of RAG and Long-Context approaches on NovelQA.** We used the tiktoken tokenizer to measure the exact token counts for (a) the questions, (b) the four answer options, and (c) the prompt template from the original paper (Wang et al., 2024a). Costs are calculated using May 2024 prices. Due to the high cost of running the evaluation, we only ran our evaluations once.

| | RAG | Long-Context |
|---|---|---|
| Total Cost | $52.80 | $99.80 |
| Accuracy | 67.89 | 67.81 |
| **RAG Specific:** | | |
| Cost of embedding 88 novels | $2.512 | - |
| Cost of embedding one novel | $0.0285 | - |
| Cost per Question | $0.0222 | - |
| **Cost per QA for a new novel** | **$0.051** | - |
| **Long-Context Specific:** | | |
| Mean prompt tokens per novel | | 690.807 |
| Total tokens of questions and options | | 110,094 |
| Total tokens (prompt + questions + options) | | 170885.016 |
| Total long-context question cost | | $1.709 |
| Long-context novel cost | | $98.09 |
| **Long-context cost per novel (single question)** | | **$1.115** |
| **Comparison:** | | |
| **Cost Ratio (Long-Context/RAG)** | $\approx 21.86$ | |

Table 7: **Examples of shortcomings in current agent evaluations stemming from inadequate benchmark standardization.** This table provides additional details on the shortcomings of agent evaluations affecting current state-of-the-art agents from five papers on the leaderboards of HumanEval and WebArena. It lists all issues arising from inadequate benchmark standardization that impact the evaluations for each agent.

| Benchmark | Paper | Evaluation issues affecting this paper |
|---|---|---|
| WebArena | Zhou et al. (2024) | • Rate limits on Reddit website impacted task completions. This affected 2/129 tasks of the Reddit subset. 
 • Autologin functionality for Reddit site was not implemented correctly (i.e., open todo comment in code) (see Appendix E.1). Caused the agent to silently fail on affected tasks. |
| | Sodhi et al. (2024) | • Evaluated on subset of benchmark. Remove 8 problems from WebArena for unclear reasons. 
 • Agent is evaluated incorrectly. Original log files provided by authors mark failed tasks as successfully solved. See Listing 2 for an example. 
 • Rate limits on Reddit website impacted task completions. This affected 30/129 tasks of the Reddit subset. See Listing 2 for an example. |
| HumanEval | Zhou et al. (2023) | • Evaluate on subset of benchmark. Authors remove four problems in their evaluations-three because HumanEval does not provide example tests and one for unclear reasons. 
 • Agent is evaluated incorrectly. Generated solutions are only evaluated on true test cases for a subset of the tasks in HumanEval. Leads to incorrect solutions passing as correct. |
| | Zhong et al. (2024) | • Underreport baseline. The paper claimed an accuracy of 75.0%, while our evaluation showed a mean accuracy of 89.6% across five runs. 
 • Evaluate on modified version of benchmark. Add example tests for 3/164 problems that are missing in the original benchmark. 
 • Claim to use GPT-3.5 for code generation using Reflexion. However, the generated program they used from the Reflexion repository relies on GPT-4 for code generation, not GPT-3.5. |
| | Shinn et al. (2023) | • Evaluate on subset of benchmark. Remove three problems missing example tests in HumanEval. Do not report this. |

Table 8: **Reported vs. reproduced accuracy of agents on HumanEval.** This table contains the reported and reproduced accuracies of the HumanEval agents part of our analysis. We report the accuracy of each agent evaluated on all 164 tasks of HumanEval. Note that insufficient standardization of the benchmark, rather than mistakes made by the agent developers, is a major contributing factor to many of the observed discrepancies. We run each agent five times and report the mean accuracy. The minimum and maximum accuracy are included in the parentheses.

| Paper | Agent | Accuracy | |
|---|---|---|---|
| | | **Reported** | **Reproduced** |
| Zhou et al. (2023) | LATS (GPT-4) | 94.4 | 88.0 (0.823-0.915) |
| | LATS (GPT-3.5) | 83.8 | 80.4 (0.787-0.835) |
| Zhong et al. (2024) | LDB (GPT-4, GPT-3.5) | 89.6 | 91.0 (0.89-0.921) |
| | LDB (Reflexion, GPT-3.5) | 95.1 | 88.9 (0.866-0.915) |
| | LDB (GPT-3.5) | 82.9 | 80.2 (0.787-0.829) |
| | GPT-4 | 75.0 | 89.6 (0.878-0.909) |
| Shinn et al. (2023) | Reflexion (GPT-4) | 91.0 | 87.8 (0.86-0.909) |

