# OpenReview forum: "AI Agents That Matter"
_TMLR — Accepted by TMLR_

### Review · Reviewer_7FMQ · 2024-10-04

**Summary Of Contributions:**

This paper identifies and addresses several critical shortcomings in current AI agent benchmarking and evaluation practices. By addressing these issues, the paper aims to enhance the real-world applicability of AI agents, moving beyond mere benchmark accuracy to foster genuine advancements in the field.

**Audience:**

Yes

**Claims And Evidence:**

Yes

**Requested Changes:**

- There is an issue with the citation in the sixth paragraph: "be enough to reproduce the results that it reports (?)."
- The fourth and fifth paragraphs need to be validated on more benchmarks.
- See weakness.

**Strengths And Weaknesses:**

# Strengths
- The article analyzes the shortcomings of existing agent benchmarks from multiple angles, such as the lack of consideration for costs, differences between testing methods and actual downstream applications, the presence of shortcuts in benchmarks, and the lack of standardization. These issues are of practical value.
- The article conducts numerous experiments to verify the problems with current benchmarking.

# Weaknesses
- Many of the issues mentioned in the article are already well-known, lacking deeper analysis and more insightful viewpoints, such as uncovering more fundamental reasons or proposing improvement solutions.
  - The article mentions "Lack of clarity on the source of performance gains," but does not further analyze what the source of performance gains is.
  - In the third section, it mentions "Jointly optimizing cost and accuracy can yield better agent designs," but does not clearly explain what a better design is. Is it suggesting that as costs increase, the improvement in accuracy decreases, and therefore an optimal cost range should be set?
- Some conclusions lack rigorous analysis and sufficient experimentation.
  - The third section only conducts experiments on HotPotQA, so it's unclear whether the experimental conclusions are universally applicable.
 - In the fourth section, it mentions "Model and downstream developers have distinct benchmarking needs," but only provides NovelQA as an example. Are there similar situations with other benchmarks?
- The article discusses the challenges of AI benchmarking from five angles, with the first three involving cost, and the other two discussing shortcuts and standardization. The classification method feels unclear, and the points are not entirely independent. For example, point 2 seems to be an extension of point 1.

---

> ### Author Response · Authors · 2025-01-28
> **Thank you for your review**
>
> We thank the reviewer for their thoughtful review. The reviewer provides valuable feedback which we incorporated into the updated manuscript.
>
> > “The article does not further analyze what the source of performance gains is.”
>
> - We highlight that many “System 2” agents rely on unverified assumptions of true capability improvements
> - Our simpler baselines often match or exceed these methods, suggesting that performance gains may hinge on simple inference scaling through, e.g., repeated sampling rather than true capability improvements.
> - We advocate for deeper, cost-controlled analyses to isolate and test each contributing factor
>
> We will clarify in the paper that further analysis is necessary and important to study in future work. We will also cite other relevant papers.
>
>
> > “it mentions "Jointly optimizing cost and accuracy can yield better agent designs," but does not clearly explain what a better design is.“
> - We define “better” in terms of accuracy and cost -- hence if an agent outperforms another in one dimension while scoring comparably in the other. On HotPotQA our joint-optimization approach achieves, e.g., very similar accuracy while being much cheaper.
> - This shows that considering both cost and accuracy simultaneously can produce more resource-efficient agents. Building optimization methods specific to this dual objective is what we propose in section 3.
> - We see this as an interesting direction for future research on more sophisticated cost-aware designs.
>
> > “The third section only conducts experiments on HotPotQA, so it's unclear whether the experimental conclusions are universally applicable.”
>
> We acknowledge that our simple joint optimization method was only showcased on HotPotQA
>
> - Our results in this section are also supported by our experiments on HumanEval and NovelQA. In both cases, we were able to design agents that are much cheaper while having comparable accuracy to the best-performing agents.
> - We explicitly encourage more work in this new design field and frame our experiments on HotPotQA as a proof of concept.
>
> > “it mentions "Model and downstream developers have distinct benchmarking needs," but only provides NovelQA as an example. Are there similar situations with other benchmarks?”
>
> Yes, similar mismatches between model-centric and downstream-focused evaluation appear in other benchmarks:
>
> - On QA benchmarks, when models are evaluated without access to an external knowledge base, results might be misleading for downstream users given that the differences between models can vanish when access to a knowledge base is provided. Making inferences for downstream decisions in the first case would then neglect the potentially much cheaper cost of inference for smaller models that only score competitively with RAG
> - Additionally, in section 4 we describe how controlling for proxies of cost such as active parameter counts when comparing dense and sparse models might be interesting for model developers and researchers but is not informative to downstream developers.
> - Our survey of 14 agent benchmarks shows that many share these inconsistent or incomplete evaluation practices that often do not state clearly what level of generality they are targeting
>
> > “discusses the challenges of AI benchmarking from five angles, with the first three involving cost, and the other two discussing shortcuts and standardization. The classification method feels unclear, and the points are not entirely independent”
>
> We appreciate the comment of the reviewer and acknowledge that the findings in our sections are connected.
> Specifically,
>
> - Sections 2, 3, and 4 talk about cost-controlled agent evaluations. They provide empirical findings of the cost of agents, motivate a new design space and argue why ignoring cost is making evaluations uninformative for downstream users,
> - Sections 5 and 6 talk about reproducibility challenges in agent evaluations. We describe why agent benchmarks often allow for shortcuts and propose a framework for benchmark design. In section 6, we discover the consequence of the lack of standardization when running evaluations and offer ideas for possible solutions which we and others address in follow-up work.
>
> These issues often intersect because agent evaluation is inherently complex. Our structure clarifies that each challenge needs a targeted solution (e.g., cost-controlled evals, robust hold-out sets, and standardized evaluations).
>
> > “The fourth and fifth paragraphs need to be validated on more benchmarks.”
>
> We acknowledge that this is a limitation of our work. Our analyses focus on a few representative tasks but could gain from additional experiments on other benchmarks.
>
> In this paper, we identify overlooked methodological issues of agent evaluations rather than exhaustively validate across every benchmark.
>
> We will clarify in the paper that the fourth and fifth paragraphs are preliminary evidence, which is a limitation, and that more evaluation is subject to future work.

---

> ### Author Response · Authors · 2025-01-28
> **Thank you for your review (cont.)**
>
> > “There is an issue with the citation in the sixth paragraph: "be enough to reproduce the results that it reports (?)."
>
> Thanks for pointing this out. Here's the citation we wanted to add is Reproducibility and Replicability in Science, ​​National Academies of Sciences, Engineering, and Medicine 2019. We will update the paper

---

### Review · Reviewer_3Reu · 2024-10-11

**Summary Of Contributions:**

This paper examines the current state of AI agent benchmarking and identifies several shortcomings that hinder the development of useful real-world agents.
Contributions:
* Propose to add “cost” to the evaluation. They demonstrate that simple baselines (retrying, model “escalation”, and “warming") can outperform “state-of-the-art” agents on coding tasks (HumanEval) while being significantly cheaper.  This underscores the need to include cost as a key metric in agent evaluation. Although this finding is not too surprising, it is valuable that someone performs and presents a systematic investigation.
* Showcase a method for “joint” optimization of accuracy and cost. Using the HotPotQA benchmark, they modify the DSPy framework to jointly optimize for both accuracy and cost, showing that it's possible to reduce cost significantly without sacrificing performance. This introduces a new dimension to agent design.
* Suggest that model and downstream (application) evaluation are often at odds in the benchmarks. The paper argues that model developers and downstream users have different benchmarking needs. Model developers prioritize accuracy improvements, while downstream users care about cost-effectiveness in real-world applications.
* Note that many benchmarks are flawed and can result in agent “overfitting”. They propose a framework for designing benchmarks with appropriate holdout sets based on the desired level of agent generality (distribution-specific, task-specific, domain-general, fully general). They analyze existing benchmarks and find that many lack appropriate holdouts. A case study with the WebArena benchmark exemplifies this issue. Generalization has been studied in a wide range of ML subareas, so this categorization is not the first of this kind, but perhaps the first in the area of developing “agents”. Therefore, it is a valuable contribution, especially together with the insights in Table 5.
* Identify inconsistencies and errors in the reproducibility of evaluations on benchmarks (WebArena and HumanEval) and call for greater standardization in evaluation practices.

Overall I believe that the paper is a valuable contribution for the community, since it explicitly identifies and names some important issues which are present in the current benchmarks, the “sota” methods, and calls for their corrections. Such contributions are always valuable as they provide more transparency into the benchmarks and methods.

**Audience:**

Yes

**Claims And Evidence:**

Yes

**Requested Changes:**

Crucial changes:
* The paper suggests that “it is possible that System 2 techniques will be useful on harder programming tasks than those represented in HumanEval such as SWE-bench”. Evaluating this would significantly strengthen the claims in the paper.
* Propose some solutions for “human-in-the-loop” evaluations.
* Propose some solutions for the “standardization” issue. The paper identifies a lack of standardization as a major problem but offers limited concrete solutions beyond calling for a standardized evaluation framework.
* Include concrete recommendations on how to address the issue of dynamic environments in agent benchmarks.

Nice-to-have changes:
* Investigate more benchmarks (from Table 5) and more agents/models (e.g. LLMs such as Gemini, Claude, etc or other open-source models). For example, do the same findings for the proposed 3 “simple” baselines hold for other LLMs too?
* Question: how come Llama-3(uncompiled) is more expensive than Llama-3 (format.instr.)?
* Question: In Section 4.1 you claim “owever, this does not represent how users would ask questions about novels in practice. Even if users have many questions about a novel, they will likely ask them individually rather than all at once. Such sequential queries would cost orders of magnitude more because the novel has to be re-processed each time.”

Do you think it's realistic that the benchmarks capture all possible real-world use cases of how users might use the models?

How do you propose to solve this aspect in current / future benchmarks?
* Missing citation: “that the code and data accompanying a paper should be enough to reproduce the results that it reports ( ?).”

**Strengths And Weaknesses:**

Strengths:
* The explicit call for incorporating cost in the evaluation protocols.
* Evaluation of several agents in terms of both accuracy and cost on the HumanEval.
* The presented 3 baselines (retry, warming, escalation) that often beat “SotA” agents.
* Pareto curve plots of accuracy and inference cost are very insightful and should be used in the future for all new proposed methods.
* Showing that “sota” agents Reflexion and LDB cost over 50% more than simple non-agentic “warming” baseline for similar accuracy.
* Finding that many reported accuracy scores were above the maximum of the 5 runs they performed.
Question: where in your paper can I find details about this claim? It would be good to put a reference to the Table/Figure after this sentence. By this I mean: reported number in the paper = X, our best score = Y.
* Findings in Table 7 on the issues affecting different papers.
* Categorization of agent benchmarks in Table 5.

Weaknesses:
* Limited scope of empirical analysis in terms of the number of benchmarks and models evaluated.
* The authors suggest the importance of human-in-the-loop evaluation, but they don’t propose how to incorporate it into agent benchmarks.
* The same holds for the call for standardized agent evaluation framework, since they do not propose a concrete solution.
* Not directly a weakness, but the claim that “escalation strategy strictly improves accuracy while costing less than half of LDB(GPT-3.5)” is somewhat misleading, since it uses more powerful models (Llama-3 and GPT-4) for “escalated” cases.
* The paper suggests that “it is possible that System 2 techniques will be useful on harder programming tasks than those represented in HumanEval such as SWE-bench”. Not investigating such tasks is a weakness that could be addressed.

---

> ### Author Response · Authors · 2025-01-28
> **Thank you for your review**
>
> We thank the reviewer for their thoughtful review. The reviewer provides valuable feedback which we incorporated into the updated manuscript.
>
> > “Limited scope of empirical analysis in terms of the number of benchmarks and models evaluated.”
>
> - We pick representative benchmarks from multiple common agent settings: Q&A, coding, and web task automation, performing in-depth analyses on HumanEval and WebArena (Sections 2 and 5), two very popular benchmarks for evaluating agents.
> - We separately analyze NovelQA to show how it can be misleading as a predictor of downstream performance (Section 4), and HotPotQA to show how jointly optimizing cost and accuracy can lead to better cost-accuracy tradeoffs (Section 3). We also include a survey of the test-set practices of 14 agent benchmarks (Table A4).
>
> > “The authors suggest the importance of human-in-the-loop evaluation, but they don’t propose how to incorporate it into agent benchmarks.”
>
> We acknowledge that section 5.2 could benefit from additional actionable suggestions on how human-in-the-loop agent evaluations can be done.
> - We do, however, touch on suggestions for human-in-the-loop benchmarks, such as providing standardized “checkpoints” during an agent’s execution where a human can intervene with feedback
> - We also point to some specific examples (e.g., https://arxiv.org/abs/2404.10952v1) of how these evaluations are conducted
> - We will add more resources of promising approaches to clarify the argument and provide more actionable insights. For example, human interaction evaluations (HIEs) (https://arxiv.org/abs/2405.10632).
>
> We will update the manuscript and make it clear that we are not proposing a concrete solution but building on the debate in the literature on this topic.
>
> > “The paper suggests that it is possible that System 2 techniques will be useful on harder programming tasks than those represented in HumanEval such as SWE-bench”
>
> We acknowledge that the focus of the paper does not lie in providing an in-depth empirical analysis of whether more complex agent scaffolds have a competitive advantage over simple agent designs with increased question difficulty
>
> We will weaken this point of the paper to clarify that this is an interesting open research question to be studied closer in future work.
>
> > “The paper identifies a lack of standardization as a major problem but offers limited concrete solutions beyond calling for a standardized evaluation framework.”
>
> - We do more than call for standardization: we advocate explicitly for the release of evaluation scripts and mention the development of an agent evaluation framework inspired by similar efforts for model evaluations such as HELM or the LM eval harness (Section 6)
> - Our Table A6 provides specific examples of evaluation pitfalls observed in practice that showcase recurring pitfalls when running evaluations.
>
> These findings have since led to the creation of open-source agent evaluation frameworks. It is an open research question whether these effectively address the issues we identify. We are leaving a more in-depth evaluation to follow-up work.
>
> > “The claim that “escalation strategy strictly improves accuracy while costing less than half of LDB (GPT-3.5)” is somewhat misleading”
>
> We thank the reviewer for this question and want to clarify that we compute cost using the actual token usage incurred by the escalation strategy using the respective pricing for each model part of the escalation chain.
>
> The method explicitly manages and reduces cost by escalating to stronger models only when necessary, which is why overall cost can remain low despite using GPT-4 for difficult tasks.
>
> We will rephrase our statement to emphasize that the cost savings arise from limiting the use of powerful models to hard problems.
>
> > “Include concrete recommendations on how to address the issue of dynamic environments in agent benchmarks.”
>
> Inspired by similar efforts in model evaluations such as HELM or the LM eval harness, we propose mitigating the effect of the dynamic agent environment and the implications for running evaluations by creating an agent eval harness
>
> It remains an open question whether such a framework for agents would address the issues we identify effectively and we leave this for future work.
>
> > “Do you think it's realistic that the benchmarks capture all possible real-world use cases of how users might use the models?”
>
> We do not claim that any single benchmark can cover every real-world scenario. Instead, our point is that, e.g., NovelQA is primarily a model-level evaluation, not a downstream evaluation.
>
> This is fundamentally a question of construct validity. When benchmarks attempt to represent real usage but ignore factors like cost or human feedback, they fail to produce robust results.
>
> Other work on agent benchmarking has tried to address this. E.g. CORE-bench (https://arxiv.org/abs/2409.11363) and Cybench (https://arxiv.org/abs/2408.08926)

---

> ### Author Response · Authors · 2025-01-28
> **Thank you for your review (cont.)**
>
> > “Missing citation: “that the code and data accompanying a paper should be enough to reproduce the results that it reports ( ?).”
>
> Thanks for pointing this out. Here's the citation we wanted to add is Reproducibility and Replicability in Science, ​​National Academies of Sciences, Engineering, and Medicine 2019. We will update the paper

---

### Review · Reviewer_LnKb · 2024-12-13

**Summary Of Contributions:**

The paper argues that current AI agent benchmarks are inadequate for evaluating agents and identifies+elaborated on key issues like the lack of appropriate hold-out sets, the absence of standardized evaluation practices, and the benefits of considering both cost and accuracy together in evaluation, etc

**Audience:**

Yes

**Claims And Evidence:**

No

**Requested Changes:**

It would be helpful if the paper could identify specifically what is its focus in the agent evaluation space and how is that different from past research in ML evaluation, and then design the paper around it.

The current title does not reflect the paper's content and might be useful to modify.

There are many instances in the paper where the writing would benefit from more clarity and rigor.

**Strengths And Weaknesses:**

While the paper raises valid concerns about the current state of AI agent benchmarking, its core arguments and recommendations are similar to existing issues and practices within the broader field of machine learning evaluation that have been discussed in depth in past literature. The need for careful selection of held-out sets that are more representative of the downstream task is a fundamental requirement in ML, not a novel insight specific to agents. Similarly, the importance of standardized evaluation procedures and reproducibility has been extensively discussed and addressed in the ML community for years.

The paper's proposed framework for choosing hold-out sets, while presented in the context of agents, builds heavily on familiar concepts of in-distribution and out-of-distribution generalization. Framing this as a new contribution specific to agents overlooks the extensive research in this space in ML such as domain adaptation.

The paper's emphasis on human-in-the-loop evaluation, while relevant, offers little concrete methodology beyond acknowledging its importance. The challenges of incorporating human feedback in evaluation, such as subjectivity and scalability, are well-known, and the paper does not contribute significantly to understanding or tackling them.

Finally, while the paper is a useful reminder of good evaluation practices, it does not provide novel insights or solutions related to AI agent benchmarking or demonstrate meaningful nuances that distinguish agent evaluation from traditional ML model evaluation. The paper's impact is therefore likely to be limited.

---

> ### Author Response · Authors · 2025-01-28
> **Thank you for your review**
>
> We thank the reviewer for their thoughtful review. We have incorporated several points of feedback into our updated manuscript.
>
> >“While the paper raises valid concerns about the current state of AI agent benchmarking, its core arguments and recommendations are similar to existing issues and practices within the broader field of machine learning evaluation.“
>
> We agree that reproducibility of evaluations is a broad issue that extends beyond agent benchmarks. We will make the paper more clear to emphasize the reproducibility issues specific to agent benchmarks. In the paper, we highlight three unique reproducibility challenges to agents:
> - First, we demonstrate how many agent benchmarks, unlike other machine learning domains, lack test sets. This can lead to overfitting during the development process and hinders reproducibility in other domains. This would be a grave error in any other ML field, so it is shocking to see it is accepted in agent benchmarks.
> - Second, we show challenges unique to reproducing agents in their environments, such as how certain environments themselves are not reproducible, such as the rate limits found in WebArena. This is not the case for prior ML benchmarks, which are usually simpler.
> - Third, we show inconsistent evaluation practices of different agent architectures are used within the same benchmarks, such as how the HumanEval agent developers rely on different subsets of the benchmark when running their evaluations.
>
> Beyond reproducibility, we make a couple of contributions specific to agents. We show why agent evaluations need to be cost-controlled to be useful for downstream users. This analysis reveals how simpler architecture often outperforms more complex “system 2” approaches that have received much attention. In section 3, we use empirical evidence on HotPotQA to motivate a new design space for agents that jointly optimize cost and accuracy.
>
> >“The paper's proposed framework for choosing hold-out sets, while presented in the context of agents, builds heavily on familiar concepts of in-distribution and out-of-distribution generalization.”
>
> We acknowledge this critique and will cite existing work on generalization and domain adoption in the ML literature.
>
> - However, agents unique and different challenges: they operate in dynamic, often multi-step environments, and real-world “downstream” deployment heavily relies on interpreting narrow benchmark gains as broadly generalizable.
> - Unlike many domain adaptation settings, agent benchmarks can lack clearly defined target distributions or static data splits, making it harder to define “in-distribution" vs. "OOD". E.g. WebArena (https://arxiv.org/abs/2307.13854)
> - Our hold-out framework is an extension, not a reinvention: we build on domain adaptation principles but adapt them to handle the broader scope and dynamic nature of agent evaluations
>
> > “The paper's emphasis on human-in-the-loop evaluation, while relevant, offers little concrete methodology beyond acknowledging its importance.”
>
> We acknowledge that section 5.2 could benefit from additional actionable suggestions on how human-in-the-loop agent evaluations can be done.
> - We do, however, touch on suggestions for human-in-the-loop benchmarks, such as providing standardized “checkpoints” during an agent’s execution where a human can intervene with feedback
> - We also point to some specific examples (e.g., https://arxiv.org/abs/2404.10952v1) of how these evaluations are conducted
> - We will add more resources of promising approaches to clarify the argument and provide more actionable insights. For example, human interaction evaluations (HIEs) (https://arxiv.org/abs/2405.10632)
> We will update the manuscript and make it clear that we are not proposing a concrete solution but building on the debate in the literature on this topic.
>
> > “It would be helpful if the paper could identify specifically what is its focus in the agent evaluation space and how is that different from past research in ML evaluation”
>
> - In section 2, we describe why cost is more than just a token count. Unlike standard LLM evaluations where per-token cost is usually a fixed metric, agent evaluations can involve repeated calls, retries, and escalating model usage. We show in Sections 2 and 3 how ignoring these dynamics can lead to inflated performance and unbounded cost.
> - Agents are one of the main ways LLMs are used for by downstream users. Benchmarks and evaluations should be sensitive to this. Yet, current evals don't appreciate the difference between model and downstream evals. Section 4 addresses this concern.
> - We show in Section 5 how current agent benchmarks are often too narrow or amenable to shortcuts, preventing us from making inferences about broad agent capability. We find that the lack of "general-purposeness" that was discussed for past waves of ML also holds for agents. In fact, it is particularly egregious for agents, because of claims made by developers to a large audience

---

> ### Author Response · Authors · 2025-01-28
> **Thank you for your review (cont.)**
>
> > “There are many instances in the paper where the writing would benefit from more clarity and rigor.”
>
> We thank the reviewer for their note. We have reviewed the draft to improve the clarity and rigor.

---

### Decision · Action_Editor_ynb4 · 2025-02-26

**Recommendation:** Accept as is

**Comment:**

# Summary of the paper
This paper focuses on AI agents in the context of LLMS. It describes the current techniques for evaluating agents, identifies several issues with them and proposes some improvements (or directions of improvements).

They first focus on cost-performance comparison, showing that the evaluations must be cost controlled, otherwise they lose their meaning. Then then propose a framework to jointly optimise cost and performance, and show that it is efficient on the HotPotQA benchmark.

Then, they focus on benchmarks, and they identify three main shortcomings of standard benchmarks. 1) Benchmarks are not always aligned with the actual applications of the model, 2) Benchmarks can be over optimized and 3) benchmarks are not always evaluated ina standard way in the literature. They illustrate their point by specifically studying WebArena and HumanEval benchmarks.


# Summary of reviews and discussion

Overall, these contributions were deemed useful to the community. Evaluating language models is a very tricky task and it is often the case that benchmarks are either not consistent between publications or do not represent the actual “performance” of the model. Thus, going towards more standardization is always beneficial.

All the reviewers agreed that the problems identified in the paper are valid and a useful contribution.

The reviewers identified two main weaknesses

1. A lack of novelty: the notion of generalization and overfitting is already known in ML.

Indeed, the second part of the paper, around benchmarks, is quite similar to very standard notions of overfitting or OOD generalization in ML. However, I agree with the author's response that the case of AI agents is specific, and is too complex to just be framed as classic OOD generalization. In particular, it is sometimes very hard to properly define  “in-distribution" vs. "OOD", as the authors point out in their response. I thus think it is worth the investigation.

2. Lack of actual solutions

The authors indeed insist on solutions involving human-in-the-loop, and point out standardization issues, but not offer concrete solutions to those issues. I agree This would make the paper even better, but, especially after the update, the authors are clear in the fact that they do not propose a solution but identify flaws, and I think it is a sufficient contribution.

The authors have correctly answered the other reviewers' suggestions.

# Conclusion

This paper tackles a very active and important subject of language models evaluation for agentic tasks. While it does not have high novelty contributions or SOTA methods, it shows important shortcomings in a key part of the literature; and at the minimum points to directions of solution.

In conclusion, as the contribution is correct to our knowledge, and useful to the ML community, it fits TMLR. I propose to accept the paper.

**Audience:**

Yes

**Claims And Evidence:**

Yes.